# Ultrasensitive detection of nucleic acids using deformed graphene channel field effect biosensors

Michael Taeyoung Hwang[1,8], Mohammad Heiranian[2,8], Yerim Kim[2,8], Seungyong You [1], Juyoung Leem[2], Amir Taqieddin [2], Vahid Faramarzi[3], Yuhang Jing[4,5], Insu Park[1], Arend M. van der Zande [1,2,6], Sungwoo Nam [2,6,7], Narayana R. Aluru[2,6✉] & Rashid Bashir[1,2,3,6,7✉]

Field-effect transistor (FET)-based biosensors allow label-free detection of biomolecules by measuring their intrinsic charges. The detection limit of these sensors is determined by the Debye screening of the charges from counter ions in solutions. Here, we use FETs with a deformed monolayer graphene channel for the detection of nucleic acids. These devices with even millimeter scale channels show an ultra-high sensitivity detection in buffer and human serum sample down to 600 zM and 20 aM, respectively, which are ~18 and ~600 nucleic acid molecules. Computational simulations reveal that the nanoscale deformations can form 'electrical hot spots' in the sensing channel which reduce the charge screening at the concave regions. Moreover, the deformed graphene could exhibit a band-gap, allowing an exponential change in the source-drain current from small numbers of charges. Collectively, these phenomena allow for ultrasensitive electronic biomolecular detection in millimeter scale structures.

[1] Holonyak Micro and Nanotechnology Laboratory, University of Illinois, Urbana, IL, USA. [2] Department of Mechanical Science and Engineering, University of Illinois, Urbana, IL, USA. [3] Department of Bioengineering, University of Illinois, Urbana, IL, United States. [4] Beckman Institute for Advanced Science and Technology, University of Illinois, Urbana, IL, USA. [5] Department of Astronautical Science and Mechanics, Harbin Institute of Technology, 150001 Harbin, Heilongjiang, P. R. China. [6] Materials Research Laboratory, University of Illinois, Urbana-Champaign, IL, USA. [7] Department of Material Science and Engineering, University of Illinois, Urbana-Champaign, IL, USA. [8] These authors contributed equally: Michael Taeyoung Hwang, Mohammad Heiranian, Yerim Kim. ✉email: aluru@illinois.edu; rbashir@illinois.edu

All-electrical detection of biomolecules and specifically nucleic acids are of great interest for gene-expression investigations[1], pharmacogenomics[2], drug discovery[3], and molecular diagnostics[4–6]. These methods also offer considerable promise for forensics[7], environmental monitoring[8] and global personalized medicine[9]. In particular, field effect transistor (FET) based detection of nucleic acids has drawn great attention as label-free and highly sensitive biomolecular sensing platform, which can be readily integrated with other electronic components, such as data analyzers and signal transducers. 2D materials, such as graphene, are attractive due to their unique properties, such as ambipolar field effect, high carrier mobility, biocompatibility, mechanical strength, and flexibility[10]. 2D materials intrinsically exhibit high sensitivity in detection of charged biomolecules due to their ultimate thinness and extremely high surface to volume ratio. Compatibility to the conventional CMOS fabrication process is another potential advantage of using 2D materials, which carbon nanotube, Si-nanowire, nanoparticles do not have. Especially, large area graphene, which is grown through chemical vapor deposition (CVD) method, has been utilized in electrical systems, such as FET device for bio-sensing[5] including detection of pH[11], microorganisms[12], blood sugar[12], and more specifically, proteins at concentrations of 10 fM[12,13], and nucleic acids (DNA or RNA) at the 100 fM concentrations[5,14]. There are a few reports that showed DNA and RNA detection at aM level, however, had significant level of background noise and lacked robust controls[15,16]. Further sensitivity would be highly desirable for detection of very rare molecules, such as micro RNA (miRNA) or cell-free DNA (cfDNA), from unamplified samples[17].

It is important to detect DNA/RNA, such as miRNA circulating in serum or plasma with high ionic strength. Such detection could enable liquid biopsy, which can replace invasive tumor-tissue biopsies in many diagnostic applications. The existing approaches to monitor cancer-related miRNA is based on the polymerase chain reaction (PCR)[18]. Unfortunately, PCR is susceptible to interference by the inhibitory factors in biological samples, therefore not suitable to analyze miRNA directly from blood or serum samples[19]. Moreover, the result can be misinterpreted by bias and artifact due to the amplification efficiency of different sequences[20]. Therefore, there is an urgent need to develop amplification- and purification- free method to directly detect miRNA from biological samples such as serum.

One of the major hurdles to lower the detection limit of FET-based biosensor is shielding of the molecule charge by the counter ions in solution (termed Debye shielding)[21,22]. Outside the Debye length, which is <1 nm in physiological solutions, the charges are electrically screened. An increase in the Debye length can result in reduced screening effect and allow for a more sensitive electrical detection of charged biomolecules. While methods have been proposed to overcome this intrinsic limitation of FET biosensors[21,22], these have focused on detecting biomolecules which are larger than the Debye length itself. None of the reports has tried to overcome the concentration limit of detection by modulating the Debye screening. Moreover, none of the works have modulated Debye screening in clinical solution such as serum or plasma[21,22].

Computational reports have also predicted that the curved morphology of sensing materials can affect the Debye length (or volume), which can increase in concave regions of a nanowire sensor[23]. Thus, we hypothesized that if the surface of the sensing channel can be curved or bent at the micro- and nano-meter scale, the Debye length could be modulated resulting in higher sensitivity. Previous works have shown that 3-dimensional "crumpled" graphene can be created by deformations at the micro- and nano-scale on pre-strained thermoplastics by relieving the stress and inducing buckle delamination of the graphene[24]. This approach can be used to engineer curving and bending of 2D materials and thin films. Several applications have been investigated using the mechanically-tunable crumpled graphene such as stretchable photosensors[25], nanoplasmonic sensor[26] and strain gauges[27], and the in-plane strain can change the electronic properties of graphene, opening a bandgap with a 1% stretch[28,29]. Moreover, crested 2D materials FETs recently showed large increase in the mobility as compared to standard devices[30]. It has also been separately shown that flat 2D semi-conducting material such as $MoS_2$ can be 10 times more sensitive than flat semi-metallic 2D graphene for biosensing applications[31]. Hence, we hypothesize that nanoscale bending of 2D graphene in 3-dimensions could result in high sensitivity due to modulation of the Debye length (or volume), and possibly due to strain induced band-gap opening in the graphene channels. Such deformed graphene (curved or bent 2D) layers have not yet been used for biosensor applications.

Here, we report the use of these deformed and bent (crumpled) graphene FET-based electrical biosensors for ultra-sensitive detection of DNA/RNA molecules down to 600 zM of limit of detection (LOD) on millimeter scale structures. To the best of our knowledge, this is the highest sensitivity reported so far using any electronic biosensor for detection of DNA. Because of the simple fabrication process, the presented approach has several benefits over structures such as nanoribbons or nanopores[32]. The process does not require electron beam lithography to fabricate the nano-confined devices. The realization of the bent and crumpled graphene is achieved by macroscopic manipulation of the 2D layer and the resulting "nano"-sized features exhibit the superior sensing performance, while allowing facile fabrication and reproducibility. We demonstrate detection of 22-mer single and double stranded molecules by adsorption and hybridization experiments, respectively. We also showed that the target miRNA (let-7b) and a DNA probe hybridization was detectable as low as 600 molecules in 50 μL of buffer and undiluted human serum directly without amplification. The performance of the sensor was further enhanced using peptide nucleic acid (PNA) probe, which showed 600 zM of LOD, ~18 molecules in an hour of incubation time. We show via molecular dynamics simulations the formation of electrical 'hot spots' at the nanoscale crumpled graphene regions where the Debye length can increase and also result in a local high potential due to the charge of DNA/RNA. Furthermore, the bending of the graphene monolayer at these hot spots can result in opening of a bandgap and provide for an exponential increase in the conductivity in vicinity of the biomolecules. These effects combined can allow for a measurable current change in millimeter scale channels even with 600 zM concentration of the target molecules.

## Results

**Characterization of deformed graphene FET biosensor.** The scheme for the graphene FET DNA sensing is illustrated in Fig. 1. Probe DNA will be anchored via a linker molecule on flat and crumpled graphene channel of FET sensors, and the target DNA will be hybridized. The concept of Debye length modulation along the flat and crumpled graphene is depicted in Fig. 1a. The dotted blue line represents the Debye length from the graphene surface. Flat graphene has a constant Debye length; however, the Debye length fluctuates at the peaks and the valleys of the crumpled graphene. The changes in Debye length on the crumpled graphene expose more of the DNA as compared to the flat graphene. Debye screening is weaker on the crumpled graphene, potentially enabling a higher sensitivity detection of DNA. The FET fabrication process is shown in Fig. 1b. The graphene

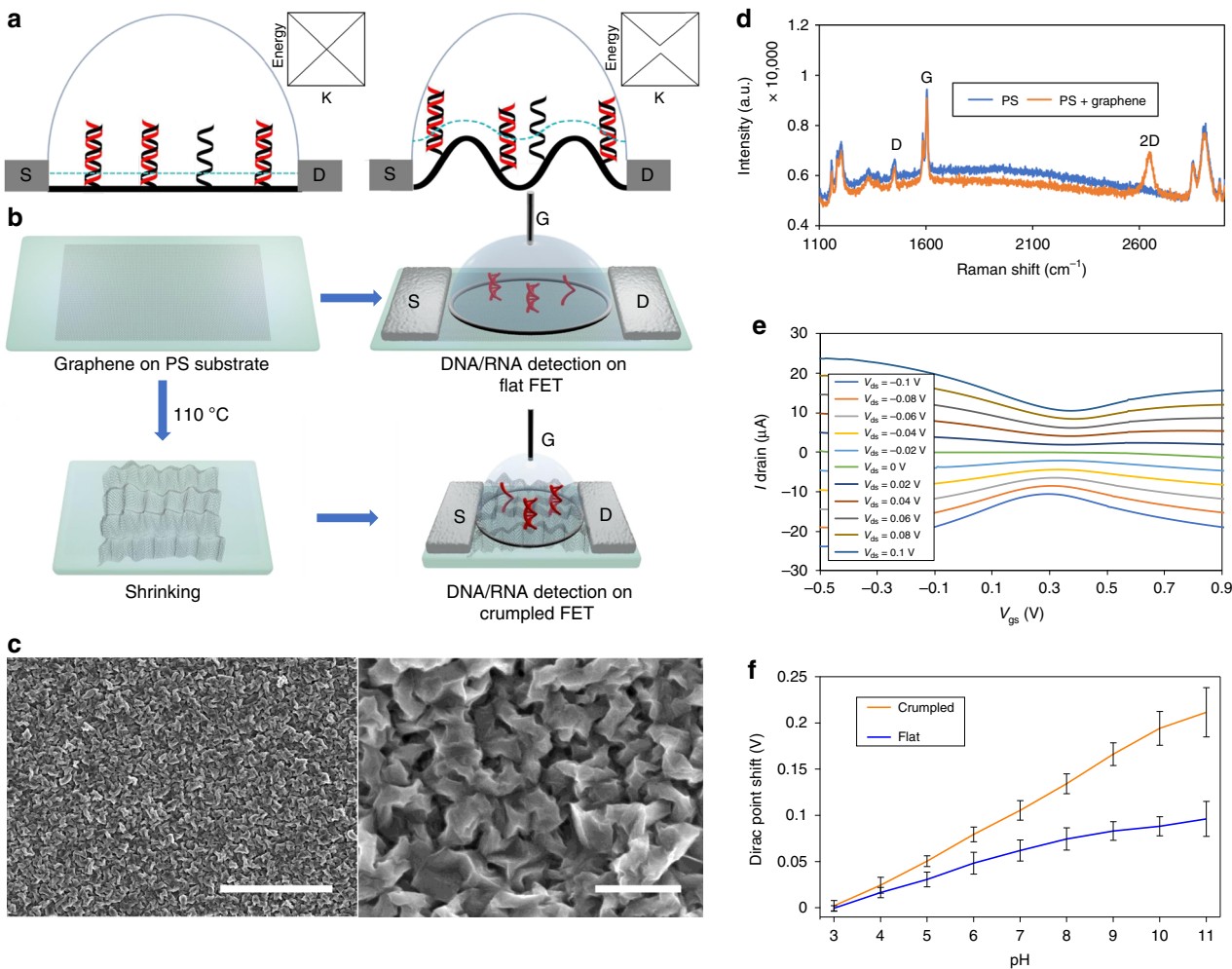

**Fig. 1 Scheme and characterization of flat and crumpled graphene FET biosensor. a** Cross-sectional scheme of the flat (left) and crumpled (right) graphene FET DNA sensor. Probe (black) and target (red) DNA strands are immobilized on the surface of graphene. The blue dot lines represent Debye length in the ionic solution and the length is increased at the convex region of the crumpled graphene, thus more area DNA is inside the Debye length, which makes the crumpled graphene more electrically susceptible to the negative charge of DNA. The inset boxes represent qualitative energy diagram in K-space. Graphene does not have intrinsic bandgap. However, crumpled graphene may open bandgap, which is discussed in the later section and supplementary table 4. **b** fabrication of FETs and experimental process flow. Graphene on pre-strained PS substrate was annealed at 110 °C to shrink the substrate and crumple the graphene. Then source and drain electrodes were applied and solution-top gate was used. In case of flat graphene FET, the annealing process was omitted. **c** SEM images of crumpled graphene. The scale bar is 5 μm (left) and 500 nm (right). **d** Raman spectroscopy of crumpled graphene and PS substrate. **e** Charge transfer characteristics of the fabricated crumpled graphene FET. $V_{gs}$ vs $I_{ds}$ (bottom) with the variation of $V_{ds}$ graphs showed shift in the Dirac point. **f** Dirac point shifts of the FET sensor plotted as a function of pH values. $n = 5$, mean ± std.

channel (1 × 15 mm) was transferred onto a polystyrene substrate using an established method[24]. To produce the crumpled FET device, the graphene on polystyrene substrate was annealed at 110 °C for 4 h. This annealing process induces the shrinkage of the underlying pre-strained thermoplastic substrate which results in buckling of graphene[24]. The optical image of the crumpled graphene device is in Supplementary Fig. 1. For the flat FET device, the annealing process was omitted. Then the source and drain metal electrodes were formed and a solution reservoir was created. When performing fluid measurements, buffer solution was placed in the reservoir in the device and a gate voltage was applied directly to the top of the buffer solution.

The morphology of the flat and crumpled graphene was characterized by scanning electron microscope (SEM) and atomic force microscope (AFM). Disorganized herringbone-like structures were observed (Fig. 1c) and the size of large wrinkles were a few microns, however, when further magnified, finer wrinkles as

small as a few hundred nanometers were observed. Debye length modulation would be attributed to those small wrinkles[24]. AFM derived topography data (Supplementary Fig. 2b) shows increase in the surface roughness for the crumpled graphene. It should be noted that the measured topography and RMS roughness values are underestimated due to the resolution of the AFM tips being larger than the ultrafine nature of the crumples. Raman spectroscopy analysis shows that the quality of graphene was intact after the crumpling process (Fig. 1d). The Raman D-to-G peak of the flat and crumpled graphene had similar intensity ratio. The background Raman spectrum of polystyrene substrate only is also shown in Fig. 1d and ref. [24] in Supplementary Fig. 3a, b. To determine if the graphene was still monolayer after the crumpling process, Very High Bond (VHB, 3 M) substrate was used and analyzed (Supplementary Note 1 and Supplementary Fig. 4). To confirm that crumpled texturing will not adversely affect graphene's electrical properties, the source-drain current

was measured, and the resistance was found to be between 8 and 12 kΩ over the measured devices. Sheet resistance of the graphene was measured by van der Pauw method and found to be ~450 Ω (Supplementary Fig. 5 and Supplementary table 1)[33].

Then the flat and crumpled graphene FET sensor were characterized with liquid gate. Graphene FET generally shows intrinsic p-type behavior due to negatively charged impurities underneath the graphene sheet which are trapped during the transfer process[34]. The conductance of the graphene channel was modulated by liquid-gate potential applied to the solution reservoir. The accumulated ions modulate the conductance of the graphene channel by either p- or n-doping effects because of the ambipolar characteristics of graphene. The ambipolar transport characteristics of the FETs are illustrated in Fig. 1e and Supplementary Fig. 3c–f. In $I_d$ vs. $V_{gs}$ curves, the Dirac points shifted as $V_{ds}$ changed. Dirac points were positioned between $V_{gd}$ of 0 and 0.5 V, which is mainly related to the work function difference of Ag/AgCl gate electrode and the graphene. The measurements were repeated over time and confirmed that the Dirac voltage values were stable in PBS buffer solution before using the devices for measurements (Supplementary Note 2 and Supplementary Fig. 6).

**Performance comparison of crumpled and flat graphene devices.** The devices were then used for pH sensing and when the H+ ion concentration changes, the current through the transistor will change accordingly (Fig. 1f). Interestingly, the crumpled graphene ISFET showed a larger shift in Dirac point from pH 3 to pH 11 as compared to the flat graphene ISFET (Supplementary Fig. 7). This might happen because the nanoscale morphology of the crumpled graphene perturbs the regular electrical double layer (EDL) and small ions could be trapped in the crumpled structure. This will be discussed later in the computational simulation section.

We then examined electrical sensing of DNA in fluid. Consistent with prior reports[5,14], we also observed that the physical contact of ssDNA strands (let-7b sequence, Supplementary Table 2) imposed n-type doping effects on flat graphene, resulting in a negative shift of Dirac point as shown in Fig. 2a–c. This is attributed to the interaction between graphene and electron-rich nucleobases in DNA molecules[35]. When the screening effect caused by ions in the buffer solution becomes strong, the charge impurity scattering caused by adsorbed DNA molecules is reduced[36]. The electrical effect of DNA on graphene gets weaker, generating smaller signals. For the flat graphene FET device, no significant shifts were observed below 2 pM concentration, whereas for higher concentrations, the Dirac point shifts were clearly observed (Fig. 2a–c). As higher concentrated DNA molecules were introduced, the IV curves gradually shifted to the left with the overall shifts of up to 80 mV. The crumpled FET device showed much large differences in the Dirac shift and even 2 aM of DNA molecules in solution resulted in IV shift of up to 20 mV and the total shift was ~180 mV (Fig. 2d–f). This corresponded to about 600 molecules in 50 μL solution added to the sensor.

Next, we investigated DNA hybridization to measure the sensitivity of the FET biosensor. Probe DNA molecules were immobilized as reported previously (Supplementary Fig. 8. and Supplementary Table 2)[5,14]. To verify that the DNA molecules were immobilized on the graphene, the surfaces of flat and crumpled graphene were probed using an AFM. The increased surface roughness was observed consistently for both flat and crumpled graphene (Supplementary Fig. 2), pointing to immobilization of the probe DNA. As shown in Fig. 2d–f, we were able to see that the Dirac point of the graphs shifted toward the left with increased concentration of the complementary DNA as low as 2 aM (let-7b, Supplementary Table 1). For the flat graphene

devices, negligible shifts were observed from 20 aM to 200 fM, and significant shifts of 2 pM or higher when the complementary DNA was introduced. The data fits well with Sips model for both crumpled and flat devices (Fig. 2g, Supplementary Note 3 and Supplementary Table 3)[6]. For the crumpled graphene device, 20 aM concentration of DNA showed a 12-mV shift. Some devices even showed a few mV of shift at 2 aM, however, these overlapped with standard deviation of negative controls. Therefore, it is reasonable to conclude that 20 aM was the limit of detection. The p-value of Dirac point shifts between crumpled and flat graphene FET biosensor at 20 aM of target DNA hybridization is shown in Supplementary Fig. 9. This indicates that the crumpled graphene FET biosensor reported here exhibits the highest sensitivity reported to date (20 aM in 50 μL, ~600 molecules) which is about 10,000 times more sensitive than prior reports from electrical biosensors[35].

We also repeated the hybridization tests using PNA probe. It has been reported that PNA probe showed one order of higher sensitivity as PNA does not have the negative charges originated from phosphate backbone of DNA. Moreover, PNA does not need addition of NH2 functional group to react with the linker (Pyrenebutanoic acid succinimidyl ester) (Supplementary Fig. 10), which reduces the distance between PNA probe and the graphene surface as compared to the DNA probe. Hence, a higher sensitivity is expected when using PNA probe[37]. Surprisingly, the LOD was improved down to 600 zM, which is ~18 molecules of DNA (Fig. 2h). Note that the total Dirac point shift is smaller than using DNA probe, as PNA test were measured in 1× PBS while 0.1× PBS was used for the DNA probe test.

Taking into account convection-diffusion-reaction considerations[38], evaporation induced convection and surface roughness effects on molecular absorption can facilitate the transport of nucleic acids to the graphene surface, reducing the diffusion-reaction time and result in high-sensitivity detection[39–41] (Supplementary Note 4 and Supplementary Fig. 11). While about 35% of initial volume was evaporated in 1 h in our experimental set up (Supplementary Fig. 1b, c, see Supplementary Note 3 for detailed explanation), target molecules could be transported along convection flows to the vicinity of the sensing surface (see Supplementary Note 4). The surface roughness of crumpled graphene may also influence the molecular reaction process, compared to a flat graphene. As the crumpled graphene forms randomly oriented valleys-and-peaks, the molecular residence time can increase by $10^2$ X as compared to the diffusion time scale[42,43] (see Supplementary Note 4 for detailed explanation).

We also performed quantification of DNA attached on graphene with radioactive labeling to see if there was a difference in the density of attached DNA between the flat and the crumpled graphene surface. The relative signals from flat and crumpled graphene were similar to conclude that this high sensitivity of the crumpled graphene FET sensor was not from a difference in density of the attached molecules (Supplementary Fig. 12).

To demonstrate the capability of realistic applications of the platform, we performed miRNA detection spiked in undiluted human serum, which is not only highly ionic but also a complex mixture of biological components. In the same testing time (1 h), we measured a clear Dirac point shift, but about half of the earlier measurements in PBS. We measured shifts at 20 aM compared to the negative control tests as shown in Fig. 2i. The concentration level of let-7b in human blood is known to be in the fM range[44] and flat graphene FET sensor is not capable of detecting it at this range. Our results demonstrate that the crumpled graphene FET shows distinct signals in the aM to fM range using direct label-free electrical detection of the miRNA molecules as an important application, and not require purification or extraction of the molecules as required by existing technologies.

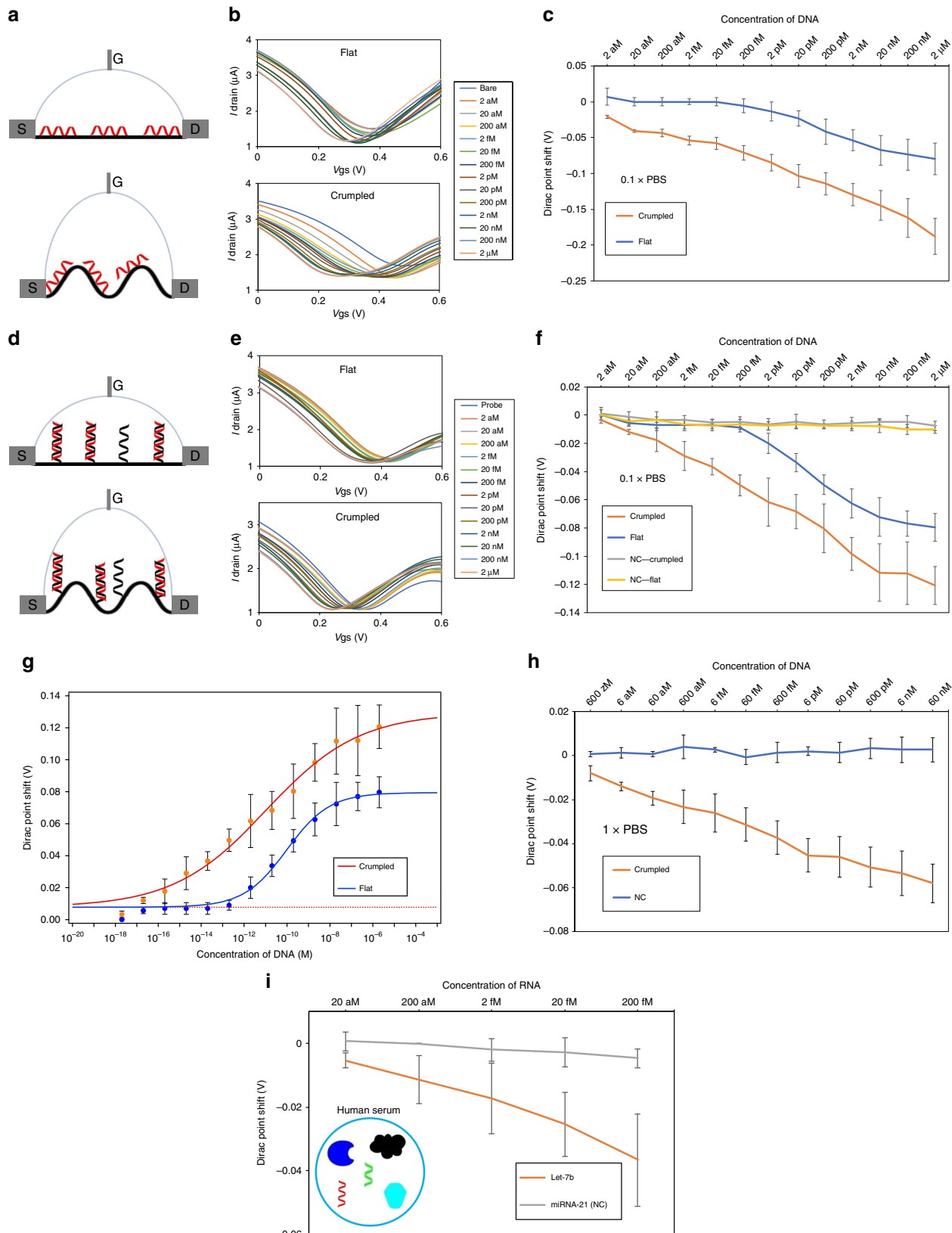

**Origin of Dirac shift**. To investigate the phenomena underlying the experimental results and the effect of ionic screening of DNA molecules, we studied the electrostatics and charge distribution of DNA and ions near flat and deformed graphene surfaces using molecular dynamics (MD) simulations (see the methods section for the simulation details). The presence of the unscreened charges (acting as dopants) carried by DNA molecules near the surface of the graphene produces long-range electrostatic potential leading to

**Fig. 2 Nucleic acids absorption and hybridization test on flat and crumpled FET. a** Lateral image of the flat (top) and crumpled (bottom) graphene FET DNA sensors. DNA (red strand) is absorbed on the graphene surface by π–π stacking. **b** I–V relationship of the flat (top) and crumpled (bottom) graphene FET sensors for the DNA absorption. DNA absorption shifted the I–V curve according to the indicated concentrations. The I–V curves shift of crumpled graphene is significantly larger than the flat device. **c** Dirac voltage shift of the FET sensor. The Dirac voltage shift is plotted as a function of the added target DNA concentration. **d** Lateral image of the flat (top) and crumpled (bottom) graphene FET DNA sensors. DNA (red strand) is hybridized with probe DNA (black strand) on the graphene surface. **e** I–V relationship of the flat (top) and crumpled (bottom) graphene FET sensors for the DNA hybridization. DNA hybridization shifted the I–V curve according to the indicated concentrations. The I–V curves shift of crumpled graphene is significantly larger than the flat device. **f** Dirac voltage shift of the FET sensor with detection of hybridization using DNA probe. NC is non-complementary control sequences used in the experiments. **g** Sips model fitting results Y-axis is absolute values of Dirac point shift. **h** Dirac voltage shift of the FET sensor with detection of hybridization using PNA probe. **i** Dirac voltage shift of the FET sensor with miRNA detection of hybridization. Target RNA spiked in human serum was treated on the FET sensor. Human serum is complex mixture of biological components. The DNA and RNA sequence used in the experiments is shown in Supplementary Table 1. All the data points are obtained from three different devices. mean ± std. *$P < 0.05$.

a change in the charge carrier density ($\Delta n$) of graphene and therefore a shift in Dirac point ($\Delta V_D$) given by refs. [45,46]

$$\Delta V_D = \frac{e\Delta n}{C_T}$$

where $C_T$ is the total gate capacitance, and $\Delta n$ is directly proportional to the charge density of the unscreened DNA molecules ($N_{DNA}^{unscreend}$) adsorbed on the graphene surface[45]. The counter-ion screening of the DNA molecules lowers the net charge of adsorbed DNA and affects the detection sensitivity. As shown in Fig. 3a–d, four different configurations are considered. In the first simulation labelled as "flat", a single-stranded DNA was equilibrated on an ideally flat graphene surface. In the three other configurations, the single-stranded DNA is adsorbed to the surface along the "concave" and "convex" regions of the crumpled graphene, and "across" the deformed graphene, respectively.

The interaction energies show that the adsorption of DNA to graphene in the concave region is the strongest. The calculated energies for the concave, convex, and across cases are $-532.187$ kcal mol$^{-1}$, $-467.484$ kcal mol$^{-1}$, and $-416.308$ kcal mol$^{-1}$, respectively (Supplementary Fig. 13). We should note that due to the complexity of actual graphene structure in experiments, it is difficult in MD simulations to calculate the exact degree of adsorption based on the simulations for a single type of crumpling (the graphene surfaces considered in the MD simulations). Therefore, we only investigated the relative degree of adsorption on different graphene surfaces for a single-stranded DNA molecule. See Supplementary Fig. 14 for more detail about DNA adsorption onto graphene surface.

The detection sensitivity of DNA molecules is defined by the degree to which the DNA molecules are screened by the ions present in the solution[23]. When the DNA molecules are isolated from the surface due to the ions present in the solution, the detection sensitivity can be significantly lower. We simulated the structure of ions and DNA relative to each other at the graphene interface for the four different configurations (Fig. 3a–d). The concentrations of ions (sodium and chloride) and the backbone of the DNA strand as well as the screening factor of ions are shown in Fig. 3e–h for the four configurations. The screening factor is then computed using the expression,

$$SF(z) = \frac{\int_0^z F([Na^+] - [Cl^-])dz}{|\sigma|}$$

where $F$ is the Faraday constant, $z$ is the normal distance from graphene surface ($z = 0$ on graphene) and $\sigma$ is the graphene surface charge density. As shown in Fig. 3a–h, because of the confined nature of the concave region, ions are excluded and are farther away from the concave graphene surface leading to increased exposure of the DNA to the graphene surface. Here, the relative position of DNA charges with respect to the ions matters.

In other words, the screening by ions starts at a larger distance away from the graphene surface in the concave case leaving much of DNA charges next to the surface unscreened. This results in weaker ionic screening of DNA molecules (or higher $N_{DNA}^{unscreend}$) adsorbed in the concave regions. Higher $N_{DNA}^{unscreend}$ induces more change in graphene carrier charge density ($n$) leading to a larger Dirac point shift. In the flat, convex and across configurations, because of weaker confinement, mobile ions are less restricted and freely present in the vicinity of the DNA molecule, screening its charge to a larger extent (lower $N_{DNA}^{unscreend}$).

The bending used in Fig. 3a–d has a wavelength of 5.41 nm and amplitude of 0.73 nm. However, a variety of crumple sizes might exist in the experiments[25]. We also modelled a narrow trench (resembling a large amplitude and short wavelength) in MD simulations with a diameter of 2.45 nm and a length of 8 nm as shown in Fig. 3i. It should be noted that the few nanometer-sized crumples are not clearly visible in SEM and AFM because of irregular feature of biaxial (uniform) crumpling process. It was reported that uniaxial crumpling is possible using the same materials and method with biaxial crumpling[24]. As a proof of principle, we created uniaxial crumple and probed the surfaces by AFM. The image in supplementary Fig. 15 shows 10 nm or smaller sized crumples. The DNA molecule sticks to the bottom of the trench and excludes the ions from that trench. The accumulation of unscreened DNA charges results in a giant local electrical potential difference between the bottom of the trench and the solution, which modifies the charge carrier density ($n$) of the graphene. Such a deep and narrow trench, referred to as an "electrical hot-spot", could provide very low ionic screening for an adsorbed DNA molecule by exposing most of DNA to the graphene surface. Therefore, $N_{DNA}^{unscreend}$ is high for such areas of the graphene and hence leads to a much higher Dirac point shift. For fM and higher concentrations of DNA, the higher Dirac point shifts for crumpled graphene observed in experiments (Fig. 2) can be explained by the fact that ionic screening is weaker and $N_{DNA}^{unscreend}$ is larger (see Supplementary Figs. 16 and 17 for details).

However, the observed Dirac point shift for aM concentrations on crumpled graphene cannot be explained merely by the low ionic screening of adsorbed DNA charges (assuming the maximum limit for $N_{DNA}^{unscreend}$). Therefore, there must be other factors that change the charge carrier density of graphene ($n$) to explain our measured detection of aM concentrations of DNA. Sarkar et al. showed that the existence of a bandgap in a single-layer $MoS_2$ leads to a higher sensitivity of charge detection compared to its graphene counterpart[31]. Here, we hypothesize the creation of a band-gap in bent graphene and next calculated the bandgap ($E_g$) for flat and crumped graphene in the absence and presence of DNA bases using density functional theory (DFT) and GW methods. Typically, GW methods are more accurate for graphene than DFT. Supplementary Table 4 shows the computed data (see the method section and Supplementary Figs. 18–22 for more details). Upon addition of DNA bases,

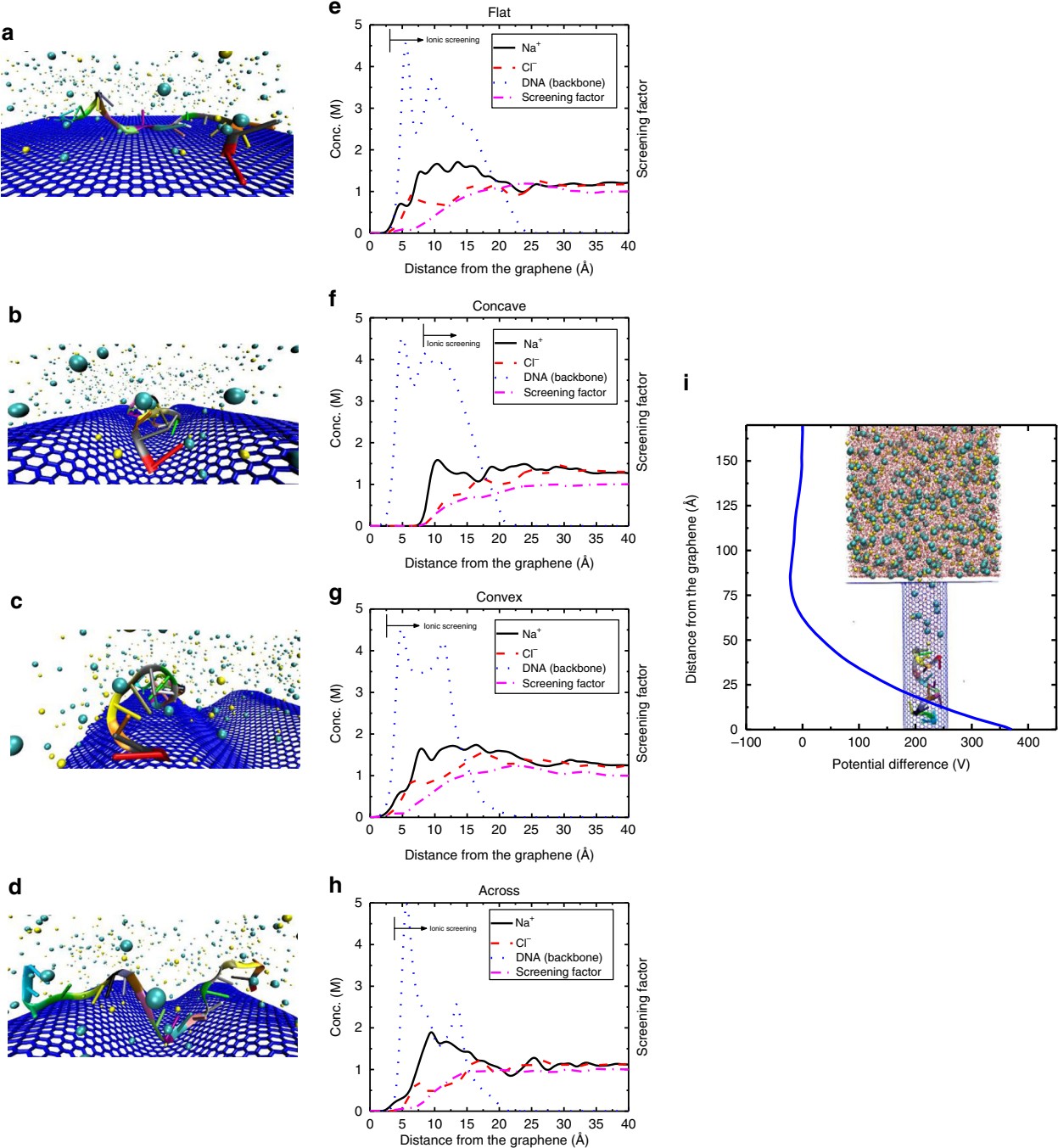

**Fig. 3 The schematic of the simulations for equilibrated DNA on. a** Flat graphene, **b** concave surface of crumpled graphene, **c** convex surface of crumpled graphene, and **d** across the graphene crumples. Graphene is shown in blue, ions are presented as cyan and yellow spheres and the DNA bases are shown in different colors. Water molecules are not shown for better presentation. The molar concentration of ions (sodium and chloride) and the backbone of DNA strand along with the screening factor of ions are plotted as a function of the distance from the graphene surface for **e** flat, **f** concave, **g** convex, and **h** across configurations of DNA. The location where the ionic screening starts to take place is shown using an arrow. In the concave region, ions are excluded due to its confinement and most of the adsorbed DNA molecule remains unscreened electrostatically. Less screening increases $N_{DNA}^{unscreend}$ and induces more charge density in graphene resulting in a larger D**i**rac point shift. **i** The 2.45-nm diameter CNT that is used to model a narrow trench in crumpled graphene, is shown with CNT and graphene carbon atoms in blue, ions in cyan and yellow, water molecules in red and DNA strand bases in different colors. The DNA adsorbs to the bottom of the trench and excludes ions near the surface (maximizing $N_{DNA}^{unscreend}$). The resulting giant electric potential modifies the carrier charge density of graphene. The potential is obtained from $V(z) = -\int\int_{z_0}^{z} \frac{q(z)}{A\varepsilon_0} dz\, dz$, where $q(z)$, $A$ and $\varepsilon_0$ are the net charge of the system (ions, DNA and water) in $z$, surface area of the bottom of the trench and vacuum dielectric constant, respectively.

there is almost no significant bandgap opening for the case of flat graphene. However, when the graphene is deformed and crumpled along the armchair direction, the bandgap opens up to 1.7641 eV by adding DNA bases. Wang et al. showed that electronic mobility ($\mu$)

in graphene decreases with increasing bandgap ($\mu \propto E_g^{-\frac{3}{2}}$) ref. [47]. As $n$ is inversely related to $\mu$, the change in $n$ (and the corresponding Dirac point shift) due to bandgap widening upon addition of DNA can be obtained[48]. To achieve the observed Dirac point shift for aM

concentrations used in our experiments, the change in bandgap from 0.4224 eV (crumpled graphene with no DNA) to 1.7641 eV (crumpled graphene with base A) must occur in at least ~$10^{-7}$% of the area of our graphene sensor (see the Supplementary Figs. 16–18). The combined observations and calculations explaining the change in carrier density due to charge exclusion and charge screening, coupled with opening of the band-gap, can collectively explain our experimentally demonstrated sensitivity of detection due to very few molecules in these millimeter scale FET sensors. (See Supplementary Note 5 for details of this section.)

**Effect of nanoscale deformation on EDL structure.** To further investigate the validity of the Debye length modulation, we measured the capacitance between the graphene and the liquid electrolyte[23]. The decreased screening for the crumpled graphene is seen once again (Fig. 4), this time in the form of decreased capacitance indicating an increased screening distance, a decreased active area, or a decreased dielectric constant. A decreased active area could be less of a factor as the total currents measured are similar across the flat and crumpled structures even in the dry state. The dielectric constant itself is not expected to change much at the crumpled graphene interface[49,50]. Hence, most of the capacitance change can be qualitatively attributed to the increased screening length. Additional sets of EDL capacitance measurement are shown in Supplementary Fig. 23. Also, different ion concentrations in the buffer solution would affect the thickness of the EDL thus resulting in a difference in the capacitance. This can also contribute to the Dirac point shift[35]. Electrical measurements were repeated with four different concentrations of PBS without the DNA molecules and the crumpled graphene showed larger shift than the flat graphene as seen in Supplementary Fig. 24. We further study the EDL length modulation in crumpled graphene in MD simulations. The molar concentration map of ions (sodium and chloride) is plotted for flat and crumpled positively charged graphene sheets in Fig. 4. No DNA molecules are present. Compared to the flat graphene, the counterions in the concave region of the crumpled graphene are distributed over a longer distance away from the surface of graphene. This is consistent with the decreased screening for the crumpled graphene in the presence of DNA molecules shown in Fig. 3.

To determine if the crumpled graphene device is capable of detecting biomolecules outside the normal Debye length at a certain buffer ionic concentration, we also varied the distance of the double strand (probe + target) DNA from the surface by a distance of 3 nt[51] (Supplementary Figs. 25 and 26). In 1× PBS buffer solution, the Debye length is ~1 nm, which is also about 3 nt long. As shown in Fig. 4h, the flat graphene device is not able to measure the 19 nt (3nt short) target DNA, however, the crumpled graphene clearly showed left shift of IV curves (Supplementary Fig. 25). The maximum shift was 40 mV and about 40% of the signal that fully complementary strand generated. From these capacitance measurements and the distance variation experiments, it is also reasonable to conclude that the EDL length increased by deforming and crumpling the graphene into the nanoscale morphology.

In future, the devices could be miniaturized to micro- or nano-sized sensor in an array format. There are many fabrication and integration process challenges towards this goal. Some of these include; (i) maintaining a high quality crumpled surface if lithography is to be performed after shrinking, (ii) performing lithography first to form smaller flat graphene islands and then performing the heating or local shrinkage to cause crumpling while keep those smaller islands attached to the underlying surface, (iii) integration of silicon FETs at each pixel for row and column addressing to form larger arrays in a silicon substrate, etc.

## Discussion

We have demonstrated nucleic acid molecule detection on crumpled graphene FET electrical biosensor with unprecedented sensitivity using DNA and PNA as a probe. DNA adsorption and hybridization tests were demonstrated using cancer-related biomarker miRNA let-7b sequence in buffer and in human serum. The limit of detection was found to be 600 zM for crumpled graphene FET biosensor and 2 pM for flat graphene. We show via experiments and simulations that the nanoscale bending and deformations increases the Debye length in ionic solution to decrease the screening of the DNA/RNA molecules, thus contributing to the dramatic enhancement of sensitivity as compared to flat graphene FET sensors. To explain the results, Molecular Dynamics simulations revealed generation of large electrical potential due to DNA molecules in the nanoscale crevices and deformed regions that exclude ions, compounded by the formation of electrical bandgap in the deformed graphene regions. These attributes coupled with increased molecular residence time due to increased roughness can result in "electrical hotspots" allowing for a change in local conductivity, hence allowing atto-molar detection of DNA in millimeter scale sensors. We demonstrate the applicability of the technology by target molecules detection in undiluted human serum for cancer-related miRNA. This technology can open opportunities for the development of more reliable and efficient diagnostic tools, and electrical point-of-care and implantable biosensors for early detection of biomolecules for various human diseases.

## Methods

**Graphene synthesis.** Monolayer graphene was grown on a copper foil via chemical vapor deposition (CVD). Before placing into the CVD furnace, copper foil was degreased with acetone and IPA, followed by nitrogen blow drying. The foil was then annealed at elevated temperature for 3 h, while 50 sccm of hydrogen ($H_2$) gas flow continuously. Monolayer graphene growth was initiated when 50 sccm hydrogen and 100 sccm methane ($CH_4$) were introduced to the CVD chamber at annealing temperature of 1030 °C, and the synthesis came to its end when two reaction gases were turned off. CVD furnace was then cooled down with flow of argon (Ar) gas, which completes the monolayer graphene synthesis process. To compare the performance among different source of graphene, graphene was also purchased from Grolltex and Graphenesquare. All the graphene showed the same LOD.

**Fabrication of graphene FET.** After graphene was synthesized, the graphene/Cu foil was spin-coated with Poly(methyl methacrylate) (PMMA). Undesired graphene formed at the back side of copper foil was removed by oxygen plasma etching. The sample was cut into 1 mm × 15 mm pieces with scissors or a razor blade. PMMA serves as a supporting layer while copper foil was etched by floating on 0.15 M sodium persulfate for about 5 h. The PMMA/graphene was rinsed by moving from the sodium persulfate solution to deionized (DI) water. The PMMA/graphene was then transferred onto a polystyrene substrate. The PMMA layer was removed by soaking in acetic acid for 5 min. The sample was annealed at 110 °C for 2 h to shrink the polystyrene substrate into ¼ of the original size and crumple the graphene. To fabricate transistor, conducting silver paste was used as source and drain electrodes at both ends of the graphene. Silicone rubber (DOWSIL™ 3140 RTV Coating) was used to insulate source and drain electrodes from liquid and construct a solution reservoir.

**Immobilization of DNA probe.** Pyrenebutanoic acid succinimidyl ester (PASE) (20 mM) in dimethyl sulfoxide (DMSO) was treated on the graphene overnight and rinsed with pure DMSO, ethanol and DI water; 50 μM of probe DNA solution was added on PASE-modified graphene for 3 h. The graphene FET with probe DNA functionalization was rinsed with 1× PBS. 200 mM ethanolamine solution was treated for 1 h to saturate the possibly unreacted amino group on PASE and rinsed with ethanol and 1× PBS solution. The volume of all treated chemicals and samples was 50 μL.

**Target DNA or RNA incubation.** The target DNA or RNA incubation was conducted by dropping complementary and negative control strands with concentrations that are indicated in the legends in Fig. 2 and incubated for 1 h in the reservoir on the graphene FET chip. Then, the chip was rinsed gently with 1× PBS. For the human serum test, the chip was incubated in serum and rinsed with serum. The serum was prepared by micro-filtering human plasma. All the volume of treated samples was 50 μL.

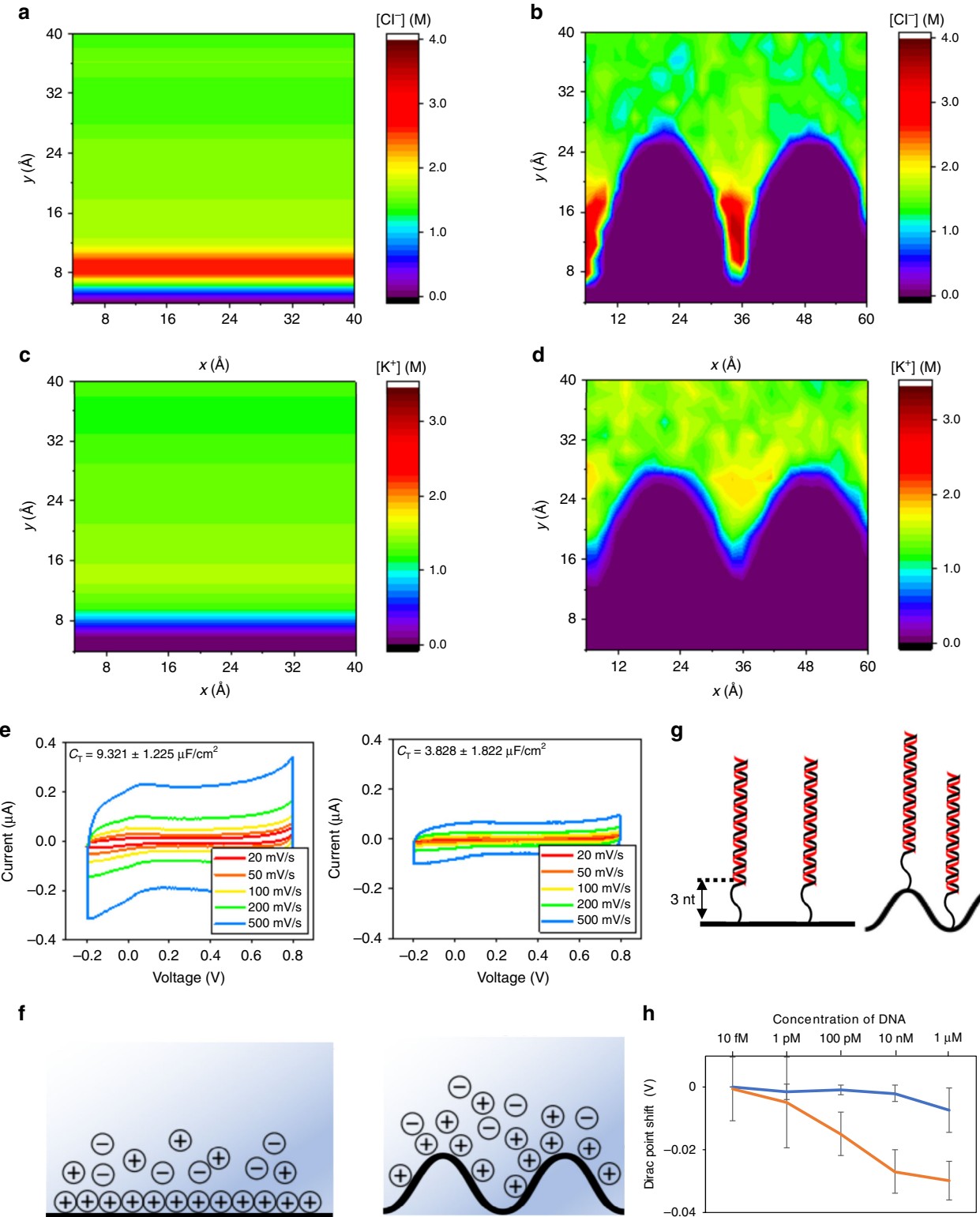

**Fig. 4 Capacitance measurement and charge layer distance effect. a–d** The molar concentration map of ions (sodium and chloride) are plotted for flat and crumpled charged graphene sheet. The counter-ions are distributed over a longer distance away from the surface of graphene in the concave region of the crumpled graphene. **e** EDL capacitance of flat and crumpled graphene. As EDL of the flat graphene is denser than the crumpled graphene, flat graphene had about three times larger capacitance than crumpled graphene. **f** ELD structures of flat (left) and crumpled graphene (right). Loosely structured EDL of crumpled graphene leads to the smaller capacitance value. **g** To determine if the crumpled graphene device has longer Debye length, distance of double strand (probe + target) DNA part was 3 nt further from the surface. **h** The flat graphene device (blue line) is not able to measure the 19 nt (3 nt short) target DNA while, the crumpled graphene (the orange line) showed left shift of IV curves. $n = 3$, mean ± std.

**Electrical measurements**. I–V curves and resistance were measured in a semiconductor parameter analyzer equipped with a probe station. Ag/AgCl electrode was used to apply gate voltage ($V_{gs}$) to the 0.1x and 1× PBS buffer solution. For the human serum test, the tests were performed in in serum. In case of DNA absorption, the graphene chips were incubated in PBS overnight because of wettability issue of hydrophobic graphene. For the serum test, the graphene chips were incubated in serum overnight and the blank measurements were repeated till there was no shift only with serum. Then target and NC RNA in serum was treated on the chip. Vg was swept from −0.5 to 1 V and drain–source voltage ($V_{ds}$) was picked between 0.03 and 0.1 V. Drain–source current ($I_{ds}$) was measured at an assigned $V_{ds}$.

**Capacitance measurements**. The capacitance measurements of flat and crumpled graphene on PS substrate were carried out using a CS 350 potentiostat (Corrtest, China) with three electrodes, including reference, counter, and working electrode. Here, silver chloride (Ag/AgCl), platinum (Pt), and the surface of the graphene channel on PS substrate were used for reference, counter, and working electrode, respectively. Cyclic voltammetry (CV) was chosen for the characterization method, while all three electrodes were immersed in 1× PBS solution for the measurement. Normalized total capacitance by measured graphene area ($C_T$) based on CV is shown in Fig. 4e.

**MD simulations methods**. Molecular dynamics simulations were performed using the LAMMPS package[52]. The systems were generated by the visual molecular dynamics (VMD)[53]. To study the effect of the crumples on the EDL formation near graphene, different simulations with different graphene surface topologies were created (Fig. 3a–d). Flat and crumpled graphene sheets were used as two different surface topologies. The crumple has a wavelength of 5.41 nm and amplitude of 0.73 nm. Each simulation box consists of a single-layer graphene sheet, a single-stranded DNA, water and ions. The DNA has 22 bases with the sequence of AACCACACAACCTACTACCTCA. The flat and crumpled graphene systems contain ~60,000 atoms with dimensions of 12.50 nm × 12.50 nm × 10 nm and 11.10 nm × 12.50 nm × 10 nm, respectively. The narrow trench system, which is modelled by a CNT with a diameter of 2.45 nm and a length of 8 nm, has ~75,000 atoms with dimensions of 10 nm × 10 nm × 17 nm. Periodic boundary conditions are applied in the x and y directions (flat graphene lies in the xy plane) for all the systems. The systems are non-periodic in z direction (normal to flat graphene plane). The Lennard-Jones (LJ) potential with a cutoff distance of 1.2 nm is used. The long-range electrostatic interactions are calculated using the PPPM[54]. SPC/E water model is employed. The SHAKE algorithm, which maintains the rigidity of each water molecule, is applied. Sodium chloride salt solution of various concentrations (1.2 and 0.6 M) is considered. The LJ parameters between atoms of graphene and water are modelled by the forcefield developed by Wu et al.[55]. The CHARMM forcefield[56] is used for the DNA strands and ions.

Before starting the equilibrium and production simulations, the energy of each system was minimized for 15,000 steps. The equilibrium simulations were then performed in NPT ensemble for 2 ns at a pressure of 1 atm and a temperature of 300 K. This ensures that the water reaches its equilibrium density. The systems were further equilibrated in NVT ensemble for another 2 ns at 300 K. Temperature was maintained at 300 K by using the Nosè-Hoover thermostat[57,58] with a time constant of 0.1 ps. The production simulations were carried out in NVT ensemble for 10 ns. Trajectories of particles were dumped every picosecond to study the structure of DNA and ions near the graphene sheet.

**DFT and GW methods**. The optimized geometry of the adsorbed DNA nucleobases on flat and crumpled graphene are obtained using DFT calculations. All DFT calculations are performed using the Vienna Ab initio simulation package (VASP)[59–61]. The local density approximation (LDA)[62] functional of Ceperley-Alder is employed based on the projector augmented wave (PAW) method[60]. The cutoff energy for the plane wave basis set is 550 eV for all calculations. All ionic positions are optimized by a conjugate gradient method until the forces on each ion are less than 0.01 eV/Å. The size of flat graphene and crumpled graphene model is about 12 Å × 12 Å and 8.9 Å × 12 Å, respectively. A vacuum separation of 40 Å between graphene and its periodic replicas is employed to eliminate the interaction between them. For accurate calculations of the electronic structures, the Brillouin zone is sampled using 18 × 18 × 1 k-point grid. The lattice parameter of flat graphene is computed to be 2.445 Å which is consistent with previous theoretical and experimental results[63,64].

Single-shot approximation of GW (G0W0)[65] was performed to obtain the band structures using VASP[59–61]. Each system has been initialized using DFT with the LDA[66] exchange-correlation functional, energy cutoff of 400 eV, and Gaussian smearing of 0.05 eV. The number of total bands is set to 512 in all structures to ensure a significant number of empty bands is available as required by GW method. The energy cutoff of the response function and the number of frequency grid points are respectively set to 90 eV and 32 to control the high memory demand by the calculation of GW. Gamma-centered k-points are selected to be 6 × 1 × 1 for structures without a DNA base and 4 × 1 × 2 for structures with a DNA base. Finally, the vacuum space is maintained large enough (>20 Å) in the periodic directions to avoid unphysical interactions between the periodic images.

## Data availability
The authors declare that the data supporting the findings of this study are included within the paper and available from the corresponding author on request. The raw data underlying Figs. 1f, 2c, 2f, 2g, 2h, 2i, 4h, S7b, S9, S12, and S24 are provided as a Source Data file.

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

## Acknowledgements

This research was primarily supported by the NSF through the University of Illinois at Urbana-Champaign Materials Research Science and Engineering Center DMR-1720633. The work was carried out in part in the Holonyak Micro and Nanotechnology Laboratory and the Material Research Laboratory Central Facilities at University of Illinois. We are grateful to our colleagues, Paolo Ferrari, Jordan Dennison, Minji Chang and Ashley Walker for help in the experimentation process as well as insights during the course of this research. We also appreciate Scott K. Silverman and Chih-Cheng Yeh for planning and performing the radioactive labeling quantification. The simulations were performed using the Extreme Science and Engineering Discovery Environment (XSEDE) (supported by National Science Foundation (NSF) Grant No. OCI1053575) and Blue Waters (supported by NSF awards OCI-0725070, ACI-1238993 and the state of Illinois, and as of December, 2019, supported by the National Geospatial-Intelligence Agency).

## Author contributions

M.T.H. and R.B conceived the work and designed the experiments. M.T.H., S.Y. and V.F. carried out the electrical measurements. M.H., A.T., and Y.J. performed the computational simulation. Y.K. and S.W.N. provided initial FET sensors and capacitance measurements. S.Y and M.T.H analyzed theoretical models of the work. M.T.H., Y.J., S.Y., and V.F. fabricated devices. J.L. conducted the AFM, SEM and Raman spectroscopy measurements. I.P. measured a part of AFM image. N.R.A. supervised the simulation efforts. R.B. supervised and led the project. A.M.v.d.Z. provided advice and a part of graphene. All authors contributed to intellectual insights of this work. M.T.H., M.H., N.R.A., and R.B. wrote the paper, and all authors edited the paper.

## Competing interests

The authors declare no competing interests.
