## [Peer Review File · Nature Communications]

Reviewers' comments:

Reviewer #1 (Remarks to the Author):

The manuscript 'Ultrasensitive detection of nucleic acids using deformed graphene channel field effect biosensors' by Hwang and co-workers describes a graphene based biosensor approach, where the graphene transducer material is crumpled and wrinkled on polystyrene substrates, which were thermally treated and collapsed to 25% of their original size. In their experiments, the authors show a previously unmatched sensitivity of short DNA and miRNA target molecules down to 600 zM.

They attribute the so far unmatched limit of charge based detection for their biosensor concept to small, concave valleys or grooves, where the ionic composition is different and the Debye screening length is modulated. They regard these areas as 'electronic hot spots' where there is a strong influence of the biomolecule charge on the transport properties and on the Dirac point of the graphene. In addition, the folding and wrinkling of graphene leads to the generation of a band gap, which is beneficial for higher sensitivity as well.

The experiments they present are very convincing. Even experiments with an important cancer related biomarker (miRNA let-7b) in human serum were successful. The biosensing experiments are framed with very detailed surface characterization and with molecular dynamic and DFT simulations.

It was a pleasure for me reading through this manuscript and I was intrigued by the detailed and thoughtful experimental planning and professional presentation of all results.

However:

To my opinion, unfortunately, some severe mistakes were done when designing the biosensor experiments. I do have several major and minor comments I like the authors to address prior to a possible acceptance of this manuscript.

Major comments:

1. I am puzzled by the concentration values presented for DNA and RNA. 600 zM to 20 aM in 50 μ l? The unit M stands usually for mol/l already. Or is it 20 amol in 50 μ l?? This would be a different value in terms of M! Please clarify this point, because this is an important issue for the sensitivity and limit of detection discussed here.
2. The fabrication protocol of the devices seems to be very easy and straightforward. I am also not so sure if silver paste is a proper contact material for graphene. This definitely should be annealed afterwards and there is a myriad of papers for selection and optimization of the graphene-metal contact. Please discuss this point and you should be at least mentioning a few of these works.
3. As with my comment 2 I understand that the fabrication process is 'handmade' and quite some variation from sensor to sensor exists. You mentioned that 5 sensors were used with 8-12 kOhm resistance over the measured devices. It seems that you used a 2-point characterization with needles contacting the silver paste. How reproducible are these experiments? For instance if you contact the same sensor several times and measure with the same liquid and DNA concentration? Please present proper controls for your concept!
4. What I do not like at all is the gradual drying out of the 50 μ l droplet during characterization. In this case the ionic strength will be increasing significantly (x1.5 !!). Does this have an influence to the Dirac point and to the experimental results? How about the other side parameters like pH and temperature? Can you completely exclude such side effects in your recordings? Are the DNA/RNA concentrations measured in a sequence or did you randomize the concentrations?
5. For all experiments, endpoint measurements are used. How is a precise timing realized, since the molecules will be dynamically binding and unbinding from the interface? If in the very low concentration levels only 60 molecules are bound on square mm surfaces to make a measureable effect, does the electronic signal already fluctuate, i.e. electrochemical 'shot noise'?
6. I am afraid that you did not use electrochemical reference electrodes to apply the gate voltage.

Only a Ag/AgCl wire was used (line 364). This is a severe mistake, since the drying out of the droplet would result in a 1.5x increase in Chloride concentration. It is of common knowledge in the field that Ag/AgCl without an ion bridge to 3M KCl will always show a response to changes of chloride concentration!

More experiments MUST be done to exclude all my concerns and possible side effects from your sensing procedure. Please present controls, stability experiments and also randomized some of the DNA/RNA sensing experiments!

Minor comments:

line 167: ... shifts of 2 pM ...

line 246: ... be noted that the ...

line 250: ... Fig. 10 shows ... (space missing)

line 319: ... increased molecular ... hotspots' allowing for a ...

In general the conclusion should be more critical also including the reproducibility of the fabrication protocol and the difficult upscale of this concept towards real clinical applications.

line 347: ... and construct a solution ...

line 364: 'Silver wire was used... ' ?? I hope this was at least a Ag/AgCl wire, but even this would not be enough due to its side sensitivity to chloride concentrations.

line 404: ... near the graphene ...

check formatting of reference 32 please.

Figure 1 caption: ... energy diagram in K-space. (please mention that this is qualitative)

Table 1: Case T: What does 'pending' mean?

Supplementary figure 2: PS and PS+graphene graphs are confused.

As a result of all my above comments, I recommend a major revision of this manuscript. I appreciate all the details of this study of characterization, sensing, simulation etc., but I am really concerned that side effects have a major contribution to the presented results! For a publication in a top journal like Nature Communications these concerns must be cleared.

Reviewer #2 (Remarks to the Author):

The report by Hwang et al. presents an ultrasensitive field-effect transistor sensor using millimeter size deformed graphene channel. The authors hypothesize that the use of deformed graphene results in the increase of the Debye length in the ionic buffer solution. To illustrate the enhanced sensitivity, similar experiments were performed on sensors with flat graphene as the control samples. The experimental results are interesting and might merit eventual publication in Nature Communications. However, the discussion on the origin of the observed phenomenon is not yet convincing and somewhat inconsistent. Moreover, there are some experimental details that are missing and would be useful for evaluation.

Below is a summary of my questions.

Questions about the sensing experiments.

- The authors have used the shift of the Dirac voltage (V_D) as the measure of the sensitivity. Please describe the experimental procedure for obtaining V_D . Specifically, mention which I_d - V_g curve was used for measuring V_D ? Was the I_d - V_g measured before and after each sensing experiment and V_D was obtained by measuring the shift between these two curves? Was the electrical characteristics of the sensor monitored continuously during each 1-hr sensing trials?
- The Dirac voltage in graphene transistors is extremely sensitive to the external factors in the surrounding environment. This is the main reason for the use of graphene as a field-effect transistor sensor. But this phenomenon is also a possible reliability issue of graphene transistor operation. Please comment on the stability of Dirac voltage of graphene transistors in your experiments? Is it a function of the voltage sweep range? Is there a hysteresis in the dual-sweep of I_d - V_g ? What considerations were made to mitigate errors due to these possible reliability issues of graphene transistors?
- The graphene transistors in this study have millimeter size. These are big dimensions for a field-effect transistor sensor. What is the rationale for choosing these dimensions? The authors note potential applications such as drug delivery. These applications require a dense array of sensors. Hence, a discussion on how sensor miniaturization might influence the limit of detection is essential.
- In Fig. 2g, the authors show the fit of the Sip model to the experimental data. What do the Sip model fits suggest about the deformed graphene vs. flat graphene sensors? Why does V_D begin to saturate at about 100 nM concentration? Are the capturing probes almost fully saturated at above this concentration? Or is there some other phenomenon taking place?
- Please provide an estimate for the density of capturing probes after functionalization for deformed and flat graphene sensors. Should the possible difference in the density of capturing probes be considered when comparing the flat and deformed graphene sensors?
- Is the deformed graphene a monolayer at every position in the channel? Or is it possible that the starting monolayer graphene might fold locally during the deformation process at different locations and form few-layer graphene? This might be verified by measuring the FWHM of the 2D peak across the sample. The Helmholtz capacitances of monolayer and few-layer graphene are different (see Nat. Commun. 5:3317, doi: 10.1038/ncomms4317).

- How was the total area estimated for calculating the area-normalized capacitance of the deformed graphene in Fig. 4e.
- Please comment on the shift of the Dirac voltage as a function of the PBS concentration (supplementary Fig. 12). Assuming that the pH of these buffer solutions was the same, the net charge in these buffer solutions is similar. Therefore, is it possible that this shift is simply due to the change of the EDL capacitance with varying the concentration of the buffer solution?

Questions about the origin of enhanced sensitivity.

- As authors stated, the shift of the Dirac voltage (V_D) can be estimated from $e \cdot n/C$, where n is the number of charges that are responsible for shifting the Dirac voltage and C is the gate capacitance. Now let us compare the flat and deformed graphene. The data in Fig. 4e suggest that the EDL capacitance of the deformed graphene is ~2.4 times smaller than the flat graphene. The data in Fig. 2e shows a Dirac voltage shift of ~80 mV for the flat graphene and a shift of ~180 mV for the deformed graphene when measuring 2 μ M DNA. The ratio of the number of charges (that cause the Dirac voltage shift) of the flat (F) to deformed (DE) graphene sensors is given by $n_F/n_{DE} = (C_{T,F} \cdot V_{D,F}) / (C_{T,DE} \cdot V_{D,DE}) = 80 \text{ mV} \cdot 2.4 / 180 \text{ mV} = 1.07$. Based on this simple calculation, the total number of charges that contributed to the shift of the Dirac voltage are almost the same for the flat and deformed graphene sensors in this example. This is inconsistent with the hypothesis that the enhanced sensitivity is due to the increase of Debye length and hence the increase of the number of charges.
- The authors suggest that the increase of Debye length cannot alone explain the observed enhanced sensitivity. Therefore, they offer another possible factor that might be responsible for the increased sensitivity, which is the bandgap opening of graphene at the functionalization sites. To justify this, they started with the Boltzmann transport theory in the diffusive regime of graphene, where the mobility is inversely proportional to the density of charged impurity scatterers (n_{imp}). Then they somehow connect this phenomenon to another one where the mobility of graphene decreases with increasing its energy gap. Please explain how these two phenomena are related? Moreover, the argument of functionalization on the armchair graphene is ambiguous. Please further elaborate. If the authors want to go in this direction, it is easy to estimate the mobility and show its relationship with n . Then they must establish how local change in the bandgap of graphene influences the transfer characteristics I_d-V_g of the device and establish this on the measured characteristics of the sensors. Note that the shift of the Dirac voltage, which is used here as the measure of sensitivity, is a macroscopic effect and is given by $e \cdot n/C$. Mobility does not appear in this relationship although it might be a function of n . However, it is unclear from the description of the manuscript how n (number of charges) in the equation for V_D is a function of mobility.

Overall, the observed enhanced sensitivity of deformed graphene is interesting. However, the discussion on the origin of this phenomenon requires further elaboration.

Reviewer #3 (Remarks to the Author):

The paper by Hwang et al. presents an experimental and theoretical study of using crumpled graphene based FET to detect nucleic acids. The authors claimed using crumpled graphene can achieve good sensitivity in DNA detection and allow fast DNA sensing in millimeter scale. Although the discovery is interesting and could open new strategies for realizing the potential of graphene (and other two-dimensional materials) in biosensing applications, the paper can not be published in the current form. I think the paper can be considered to be published in Nature Communications if the authors can address the following concerns well.

Overall, the paper is well written with systematic experimental and theoretical investigations. However, a central question to this paper is: can this method achieve fast and reliable base-specific detection of DNA? This question is a key challenge in this field. Throughout the paper the authors claim to have found a ultrasensitive sensing method and conclude that crumpled graphene have higher sensitivity than other 2D material counterparts, but can the electric signals generated by this approach be base-specific (i.e. each nucleobase on graphene FETs leads to distinct electronic responses)? In the theoretical part (especially in Table I), I notice that the band gap opening on graphene is almost similar for all nucleobases. The authors should address this point to justify the novelty they present in this study.

I also have some other questions and some minor comments on the paper:

1. The morphology of crumpled graphene is quite complicated with various concave, convex and confined regions. My concern is how stable the structure is and whether the process can be reproduced conveniently. Can author explain how they determine the structure is still crumpled "monolayer" graphene after the process?
2. The "electric hotspots" resulting from deformed graphene structure is expected. This could also be done with other confined setups such as graphene nanopores. The author could review these methods and address why using crumpled graphene is better.
3. The authors present the comparison among different adsorption scenarios of nucleobases on graphene in Figure 3. However, the presentation in (e-h) is confusing. The authors state that concave adsorption leads to less electrical screening of DNA molecules due to strong contact with underlying graphene. How is this statement reflected in Figure 3 (e) - (h)? It seems to me the screen factor curve that the author used to quantify the sensitivity for all cases in Figure 3(e) -(h) is similar to each other. I can not conclude that the concave surface (Figure 3(f)) results in superior sensitivity than other cases. Authors need to clarify this and revise the graphical presentation of their MD results.
4. The authors only consider single stranded DNA in MD. What would be the response on double stranded DNA? double stranded DNA could lead to significant difference compared with single stranded DNA (see Kabelac et al. Phys. Chem. Chem. Phys. 2012, 14, 4217-4229).
5. The authors list the comparison between the band gap of flat and crumpled graphene upon adsorption of nucleobases in different orientations. However, what is the exact orientation of nucleobases on graphene considered in DFT calculations? Authors should provide some explanations on this.
6. The electronic signals of graphene can be significantly altered by changing the local contact between graphene and nucleobases (see Ahmed et al. Nano Lett., 2012, 12, 927-931, Yin et al., J. Phys. Chem. Lett. 2017, 8, 3087-3094 and Caridad et al. , Carbon, 2018, 129, 321-334). When considering modelling the actual contact between DNA nucleobases and flat graphene/crumpled graphene, how did authors determine the three representative cases? The authors need to provide

more information of how the graphene/nucleobase system is modelled. Moreover, is the nucleobase weakly adsorbed on graphene via vdW forces? The authors do not mention whether they have used vdW corrections in DFT calculations.

7. Another concern is the missing of data in Table I. GW results are only given for some cases.

8. Among available data in Table I, I notice a great difference between band gap of crumpled graphene upon adsorption of Adenine in Orientation I and II (1.21 eV) via GW, while the band gap difference is only less than 1 meV with DFT-LDA? I wonder if this result is reasonable. From the authors' results, the GW results seem to only increase the band gap magnitude for all cases, not resulting in big band gap differences between different orientations. Authors need to clarify this.

Minor comments:

9. Figure 1(e): Is this the charge transfer characteristic of crumpled graphene FET (same to Supplementary Figure 2(f))? Authors should make this clear in the main text and Figure caption.

10. When discussing the graphene-based DNA sensing and sequencing, the authors can use this well-written review: Heerema et al., *Nature Nanotech.*, 2016, 11, 127-136.

Reviewers' comments:

Reviewer #1 (Remarks to the Author):

The manuscript 'Ultrasensitive detection of nucleic acids using deformed graphene channel field effect biosensors' by Hwang and co-workers describes a graphene based biosensor approach, where the graphene transducer material is crumpled and wrinkled on polystyrene substrates, which were thermally treated and collapsed to 25% of their original size. In their experiments, the authors show a previously unmatched sensitivity of short DNA and miRNA target molecules down to 600 zM.

They attribute the so far unmatched limit of charge based detection for their biosensor concept to small, concave valleys or grooves, where the ionic composition is different and the Debye screening length is modulated. They regard these areas as 'electronic hot spots' where there is a strong influence of the biomolecule charge on the transport properties and on the Dirac point of the graphene. In addition, the folding and wrinkling of graphene leads to the generation of a band gap, which is beneficial for higher sensitivity as well.

The experiments they present are very convincing. Even experiments with an important cancer related biomarker (miRNA let-7b) in human serum were successful. The biosensing experiments are framed with very detailed surface characterization and with molecular dynamic and DFT simulations.

It was a pleasure for me reading through this manuscript and I was intrigued by the detailed and thoughtful experimental planning and professional presentation of all results.

Response: we appreciate these positive and encouraging comments very much.

However:

To my opinion, unfortunately, some severe mistakes were done when designing the biosensor experiments. I do have several major and minor comments I like the authors to address prior to a possible acceptance of this manuscript.

Major comments:

1. I am puzzled by the concentration values presented for DNA and RNA. 600 zM to 20 aM in 50 μ l? The unit M stands usually for mol/l already. Or is it 20 amol in 50 μ l?? This would be a different value in terms of M! Please clarify this point, because this is an important issue for the sensitivity and limit of detection discussed here.

Response: We thank you for the comment and we have clarified this further. Yes 'zM' is mol/liter. Therefore 600 zM is $600 \times 6 \times 10$ molecules in 1 liter so ~ 18 molecules in 50 μ l. We have clarified this further in the manuscript.

2. The fabrication protocol of the devices seems to be very easy and straightforward. I am also not so sure if silver paste is a proper contact material for graphene. This definitely should be

annealed afterwards and there is a myriad of papers for selection and optimization of the graphene-metal contact. Please discuss this point and you should be at least mentioning a few of these works.

Response: Thank you for pointing out our missing discussion on the source/drain electrodes. There are several important works on DNA detection on graphene FET reported that have used these large area silver paste as source/drain contact even without annealing (Adv. Mater. 2010, 22: 1649-1653; Adv. Funct. Mater. 2013, 23: 2301-2307; PNAS June 28, 2016 113, 26, 7088-7093). As the measured resistance of the devices was $\sim 10\text{ K}\Omega$, and consistent with the sheet resistivity values from literature, our mm to cm scale sensors formed with the silver paste do form ohmic contacts and do not create high contact resistance, as also shown in the Supplementary Fig. 3c and 3d.

We also now added the device stability data in Supplementary Fig. 4 which further corroborates the low contact resistance devices.

3. As with my comment 2 I understand that the fabrication process is 'handmade' and quite some variation from sensor to sensor exists. You mentioned that 5 sensors were used with 8-12 kOhm resistance over the measured devices. It seems that you used a 2-point characterization with needles contacting the silver paste. How reproducible are these experiments? For instance if you contact the same sensor several times and measure with the same liquid and DNA concentration? Please present proper controls for your concept!

Response: We thank you for these questions. The simple fabrication process is a strength of the proposed approach and one of the key claims of the paper is that we can achieve the very high sensitivity with the simple fabrication. Yes, we did 2-point characterization with needles contacting the silver paste. We have checked the resistance of the graphene, before and after applying the silver paste and the resistance value was always the same. As mentioned above, there are several works with silver paste, and they all reported consistent results. When we repeat the IV measurements on our DNA sensor, the Dirac points do not change among measurements to within $\pm 2\text{ mV}$). These result has been added as stability test result in the Supplementary Fig. 4.

For the comment on the use of 5 sensors for the 8-12 kOhms resistance, these 5 devices were used to report the sheet resistivity and not used for all the measurements. We used many more devices to perform all the reported measurements.

We additionally measured the sheet resistance of our graphene by van der Paw method as shown below.

	Constant Current (100 μ A, 1 to 2 and 1 to 3))
Measured Voltage 1 to 2	8~10 mV
Measured Voltage 1 to 3	8~10 mV

Using the equation from reference paper (Chin. Phys. B Vol.26, No.6 (2017) 066801),

$$1/\sigma_{\square} = \frac{\pi}{\ln 2} \frac{R_{I1.V23} + R_{I2.V34}}{2}$$

The sheet resistance is $\sim 450 \Omega$, which is in concordance with known values. The size of the graphene in our device is $\sim 1 \text{ mm} \times 10 \text{ mm}$ and the resistance was 8~12k Ω . Therefore, the contact resistance is $\sim 3 \text{ k}\Omega$, which is in the range of normal value for the metal contact. Hence, the contact resistance of our hand-drawn silver paste contacts is in the normal range. Most importantly, we would like to point out that the nucleic acids detection data was performed and analyzed by the Dirac point shift, which is not a function of the contact resistances of the devices.

4. What I do not like at all is the gradual drying out of the 50 μ l droplet during characterization. In this case the ionic strength will be increasing significantly (x1.5 !!). Does this have an influence to the Dirac point and to the experimental results? How about the other side parameters like pH and temperature? Can you completely exclude such side effects in your recordings? Are the DNA/RNA concentrations measured in a sequence or did you randomize the concentrations?

Response: We appreciate these comments from the reviewer. Yes, ionic strength obviously affect the Dirac point, which we showed in the original Supplementary Fig. 14. The mention of drying was to ensure that we provide all experiment details and our observations during the measurement protocol and the 1.5X was an extreme case. The evaporation was not consistent from experiment to experiment. Our sensors and measurements were robust given these real experimental variabilities. In total, we performed hundreds of measurements, counting all the various experimental splits. The devices were rinsed with fresh buffer solution before every DNA measurement. We believe

that the changes that were measured were not due to change in ionic strength changes. Very importantly, the negative controls always showed stable negative results, as shown in Fig. 2f and 2h and the control and stability measurements. Hence, we do not believe that the possible reducing of the volume affects the results adversely. The reduction of volume 50% would only increase the concentration by 1.5X and not by orders of magnitudes. Hence, the sensitivity measured is not due to drying or any small reduction in volume. Rather it shows the robustness of the overall trends and demonstration of the sensors to measure the very low concentrations with a relatively simple and 'low tech' fabrication process – with the high crumpling and bending, being the key innovation that helps us realize the very high sensitivity.

5. For all experiments, endpoint measurements are used. How is a precise timing realized, since the molecules will be dynamically binding and unbinding from the interface? If in the very low concentration levels only 60 molecules are bound on square mm surfaces to make a measurable effect, does the electronic signal already fluctuate, i.e. electrochemical 'shot noise'?

Response: This is a very good comment and we very much agree that any noise effect should be very carefully considered for the detection of such a low number of molecules. In our measurements, each data point was obtained by repeatedly measuring 6 measurements every two minutes, and to make sure that Dirac point is same at least for the last two measurements. Once the Dirac point was stable, it remained stable over multiple repeats of the measurements on that device. Now we have included a new data set of Dirac point stability test in Supplementary Fig. 4, which shows stable Dirac points over more than 10 repeats of measurements. This was the criteria for choosing a good device versus a device that was not stable. We should have included this measurement before in our earlier submission. Also, please note that we have thoroughly performed negative control tests, as shown in Fig. 2f, 2h and 2i and the resulting signals were much smaller or negligible compared to the signal generated by the target nucleic acids.

6. I am afraid that you did not use electrochemical reference electrodes to apply the gate voltage. Only a Ag/AgCl wire was used (line 364). This is a severe mistake, since the drying out of the droplet would result in a 1.5x increase in Chloride concentration. It is of common knowledge in the field that Ag/AgCl without an ion bridge to 3M KCl will always show a response to changes of chloride concentration!

Response: We appreciate the reviewer's constructive criticism on this matter. Yes we used Ag/AgCl wire electrode (Warner Instruments - also, ACS Nano 2012, 6, 7, 5972-5979, Sensors and Actuators B, 2017, 250, 100–111). The Ag/AgCl wire is known to be used as an electrochemical electrode. As we have mentioned in response to comment 4 and 5, the device was rinsed with fresh PBS and we repeated the measurements till the Dirac point was stabilized. After the stabilization, Dirac point did not shift, which means that ion balance was achieved. More details are added in the manuscript.

More experiments MUST be done to exclude all my concerns and possible side effects from your sensing procedure. Please present controls, stability experiments and also randomized some of the DNA/RNA sensing experiments!

Response: We understand and appreciate the concerns and we hope that we have addressed them in the comments above and in the revised manuscript which includes additional experiments.

Minor comments:

line 167: ... shifts of 2 pM ...

line 246: ... be noted that the ...

line 250: ... Fig. 10 shows ... (space missing)

line 319: ... increased molecular ... hotspots' allowing for a ...

In general, the conclusion should be more critical also including the reproducibility of the fabrication protocol and the difficult upscale of this concept towards real clinical applications.

Response: We have added comments on the upscale of the technology to arrays and the associated challenges. Also, we had already include measurements in SCF with spiked molecules as a strong proof of concept for clinical applications.

line 347: ... and construct a solution ...

line 364: 'Silver wire was used... ' ?? I hope this was at least a Ag/AgCl wire, but even this would not be enough due to its side sensitivity to chloride concentrations.

Response: Yes this was a Ag/AgCl wire electrode. Our apologies.

line 404: ... near the graphene ...

check formatting of reference 32 please.

Figure 1 caption: ... energy diagram in K-space. (please mention that this is qualitative)

Table 1: Case T: What does 'pending' mean?

Supplementary figure 2: PS and PS+graphene graphs are confused.

Response: We truly appreciate the reviewer for pointing out these issues. We have addressed all these concerns.

As a result of all my above comments, I recommend a major revision of this manuscript. I appreciate all the details of this study of characterization, sensing, simulation etc., but I am really concerned that side effects have a major contribution to the presented results! For a publication in a top journal like Nature Communications these concerns must be cleared.

Response: We truly appreciate the reviewer for pointing out these major and minor issues. We believe that we have addressed all these concerns.

Reviewer #2 (Remarks to the Author):

The report by Hwang et al. presents an ultrasensitive field-effect transistor sensor using millimeter size deformed graphene channel. The authors hypothesize that the use of deformed graphene results in the increase of the Debye length in the ionic buffer solution. To illustrate the enhanced sensitivity, similar experiments were performed on sensors with flat graphene as the control samples. The experimental results are interesting and might merit eventual publication in Nature Communications. However, the discussion on the origin of the observed phenomenon is not yet convincing and somewhat inconsistent. Moreover, there are some experimental details that are missing and would be useful for evaluation.

Below is a summary of my questions.

Questions about the sensing experiments.

- The authors have used the shift of the Dirac voltage (DVD) as the measure of the sensitivity. Please describe the experimental procedure for obtaining DVD. Specifically, mention which Id-Vg curve was used for measuring DVD? Was the Id-Vg measured before and after each sensing experiment and DVD was obtained by measuring the shift between these two curves? Was the electrical characteristics of the sensor monitored continuously during each 1-hr sensing trials?

Response: We obtained the Dirac Voltage shift by measuring the current versus voltage curve when a solution with no molecule was introduced and then compared the current versus voltage curve to when a solution with the target molecules was introduced on the device. We performed end-point measurements and made sure that the Dirac Voltage shift was stable without adding solution with the target molecules. We describe the detailed measurement protocol in the next comment and have also included that in the supplementary information. We should have done that before and our apologies for not doing so.

- The Dirac voltage in graphene transistors is extremely sensitive to the external factors in the surrounding environment. This is the main reason for the use of graphene as a field effect transistor sensor. But this phenomenon is also a possible reliability issue of graphene transistor operation. Please comment on the stability of Dirac voltage of graphene transistors in your experiments? Is it a function of the voltage sweep range? Is there a hysteresis in the dual-sweep of Id-Vg? What considerations were made to mitigate errors due to these possible reliability issues of graphene transistors?

Response: We appreciate this comment from the reviewer. We repeated the measurement 6 times every two minutes for each data point and made sure that Dirac point has stabilized and that it is the same for the last two measurements. The device was rinsed with fresh PBS every 3 measurements. Once the device and the Dirac point was stable, the Dirac points are the same over many repeats of measurements. Now we have included a new data set of Dirac point stability test in Supplementary Fig. 4, which shows stable Dirac points for 10 repeats of measurements. Also, please note that we have extensively performed negative control tests, shown in Fig. 2f, 2h and 2i, and the signals were much smaller or negligible compared to the signal generated by the target nucleic acids.

- The graphene transistors in this study have millimeter size. These are big dimensions for a field-effect transistor sensor. What is the rationale for choosing these dimensions? The authors note potential applications such as drug delivery. These applications require a dense array of sensors. Hence, a discussion on how sensor miniaturization might influence the limit of detection is essential.

Response: This is very good comment. While we agree that future devices should be miniaturized and used in an array format, this work was the first demonstration of the use of wrinkled and crumpled graphene devices for high sensitivity detection of biomolecules. Our millimeter-sized device were prototypes for these proof-of-concept experiments that this device can detect very small number of nucleic acids, even on these large-area graphene. Fact is that our plans were to test these larger devices and depending on the outcome, work on scaling these downs. Our results were positively and surprisingly unexpected. Also, when one type of molecule or biomarker is to be detected, then these dimensions are acceptable and even preferred. We would like to note that the commercially available glucose electrochemical sensors, or the electrochemical sensors in the iSTAT cartridges, are mm to cm scale sensors measurement for single analytes. The fact that we can measure the low number (high sensitivity) of molecules even with these large sized devices is actually a major advantage and strength of this approach and the manuscript.

We also believe that further miniaturization would have to also be coupled with formation of the smaller sensor in an array format. We have already put much thought into this useful direction. Some of the fabrication and process integration challenges towards this goal would include;

- (i) maintaining a high quality crumpled surface if lithography is to be performed after the crumpling process,
- (ii) Alternatively, performing lithography first to form smaller flat graphene islands and then performing the heating or local shrinkage to cause crumpling while keeping those smaller islands attached to the underlying surface,
- (iii) Whether the metallization and contacting of each sensor should be before or after they have been crumpled,
- (iv) isolation of the metal electrodes from the fluid if the fluid is introduced on the array,
- (v) possible integration of silicon FETs at each pixel for row and column addressing in a larger array, and in this case, the use of silicon substrate, etc.

We believe that these challenges should be the subject of a follow on papers.

- In Fig. 2g, the authors show the fit of the Sips model to the experimental data. What do the Sips model fits suggest about the deformed graphene vs. flat graphene sensors? Why does DVD begin to saturate at about 100 nM concentration? Are the capturing probes almost fully saturated at above this concentration? or is there some other phenomenon taking place?

Response: Thank you for this comment. The Sips model is commonly applied to describe the statistical distribution of the molecular (or gas) adsorption energies on a solid surface.

$$|\Delta V_D(V)| = A \frac{(C/K_a)^a}{1 + (C/K_a)^a}$$

	crumpled	flat
A	0.122 ± 0.007 V	0.072 ± 0.002 V
a	0.200 ± 0.021	0.436 ± 0.053
K_a	$1.12e-11 \pm 9.44e-12$ M	$9.71e-11 \pm 3.37e-11$ M

Here;

- A is the saturation value of Dirac point voltage.
- a is association constant that characterizes the energy distribution of the DNA adsorption isotherm on the surface, which is in range of 0~1. It is in the similar range between the crumpled and the flat.
- K_a is dissociation constant. It is strongly related to the length of the binding DNA. The K_a value decreases exponentially as the adsorbed molecular size increases. We find a difference in K_a value between the crumpled and the flat surfaces.

The above implies that the crumpling feature may affect the dissociation constant of DNA. This is consistent from an earlier publication in our lab (Advanced Functional Materials, 2011, 21(6):1040-1050) which demonstrated that immobilized protein on the nano-crystalline diamond substrate with higher roughness showed stable binding for 2 weeks while the binding was less stable on the smoother carbon substrate.

In regards to the saturation of the curves, we believe that the available probes fully captured the target molecules. Prior work has shown (figures below) DNA detection with similar level of graphene FET device size and sample volume showed the similar fitting graph and saturation point at the similar DNA concentration (ACS Nano, 2016, 10, 9, 8700-8704). In their work, the 52-arrayed graphene FET was ~ 1 cm \times 1 cm and the target DNA sample volume was ~ 1 μ l (according to the author). In their data (shown below), 22 nt of DNA, which is same length as our target DNA and showed the saturation point at ~ 1 μ M - consistent with our results. Also, their probe functionalization strategy was same with ours. We assume that the surface of the graphene was fully covered by PASE linker molecule. However, not all PASE molecules would react with probe molecules as DNA is negatively charged and repulsive to each other. Thus there would be a certain distance among probe DNA molecules. PNA probe are expected to show a better sensitivity than DNA probe as PNA molecules do not have a charge and could be packed tighter on the surface as

compared to DNA.

- Please provide an estimate for the density of capturing probes after functionalization for deformed and flat graphene sensors. Should the possible difference in the density of capturing probes considered when comparing the flat and deformed graphene sensors?

Response: We thank you for this comment. In previous reports (e.g. ACS Nano, 2016, 10, 9, 8700-8704) which used the same protocol as ours, the density of the probe was estimated as $\sim 1.3 \times 10^3 \mu\text{m}^{-2}$. We expect the density would be similar level for our case. In the original manuscript, we performed an estimate of the quantification of the DNA molecules on the crumpled and flat graphene using radioactive probes and the data is shown in Supplementary Fig. 9.

- Is the deformed graphene a monolayer at every position in the channel? Or is it possible that the starting monolayer graphene might fold locally during the deformation process at different locations and form few-layer graphene? This might be verified by measuring the FWHM of the 2D peak across the sample. The Helmholtz capacitances of monolayer and few-layer graphene are different (see Nat. Commun. 5:3317, doi: 10.1038/ncomms4317).

Response: This is a good point. As shown in Fig. 1d and Supplementary Fig. 3a-b, we have performed Raman spectroscopy to characterize the graphene layers. However, as polystyrene (PS) substrate has a Raman peak close to graphene G-peak, the Raman peak marked as “G” in the plot is actually ‘graphene G peak + PS peak’. To observe the G peak without the peak from PS, we prepared flat and crumpled samples on VHB substrate (because VHB does not have Raman peak close to graphene G peak). The following was the sample preparation protocol;

1. Transfer graphene on a PDMS stamp
2. Transfer graphene onto VHB substrate
 - For Flat sample: VHB tape was put on a slide glass because the soft VHB made the contact printing process harder.
 - For crumpled sample: 100% pre-strain was applied in x- and y-axes to apply same amount of pre-strain as crumpled graphene on PS substrate.

The result graphs are below;

Based on the 2D/G intensity ratio (d), both crumpled and flat samples showed larger than 2. FWHM of 2D peak was analyzed, and in both samples, FWHM of 2D peak indicates monolayer graphene. (For monolayer graphene, FWHM < 30cm⁻¹, Nano Lett. 2007, 7, 2, 238-242). VHB has a Raman peak at ~2670 cm⁻¹ on the crumpled sample, and it overlapped with graphene's 2D peak. That would have made crumpled graphene sample's FWHM value larger than flat sample. 2D peak center was also analyzed to be larger from crumpled sample (2636.4 cm⁻¹ vs. 2628.2 cm⁻¹). Hence, we believe that based on our Raman spectroscopy analysis, both crumpled and flat graphene samples were monolayer. The reviewer is correct that since the crumpled graphene is a three-dimensional structure, out-of-plane structures may interact or even touch each other. However, the spectral analysis does not indicate apparent evidence of local folding/touching/interacting of graphene structures in the measurements we have performed and reported. We have

added this result in the supplementary Note. Also, there is a previous report that the graphene remained monolayer after the same crumpling process (Adv. Mater. 2016, 28: 4639-4645).

- How was the total area estimated for calculating the area-normalized capacitance of the deformed graphene in Fig. 4e.

Response: For all samples, they were cut and measured at 1 mm × 14 mm and used before crumpling.

- Please comment on the shift of the Dirac voltage as a function of the PBS concentration (supplementary Fig. 12). Assuming that the pH of these buffer solutions was the same, the net charge in these buffer solutions is similar. Therefore, is it possible that this shift is simply due to the change of the EDL capacitance with varying the concentration of the buffer solution?

Response: As the reviewer mentioned, the pH of the different ion concentrations of PBS buffer is same (~7.4), thus the net charge is also same. The thickness of EDL is different among those buffer solutions and this contributes to the difference in capacitance. This change in EDL thickness, results in a Dirac shift. Similar results on flat graphene has been reported earlier (Biosensors and Bioelectronics Volume 41, 15 March 2013, Pages 103-109). As seen in Supplementary Fig. 12, we have found that the crumpled graphene showed larger shift from this capacitance change as compared to flat graphene. We have further elaborated on this in the manuscript.

Questions about the origin of enhanced sensitivity.

- As authors stated, the shift of the Dirac voltage (DVD) can be estimated from $e\Delta n/CT$, where Δn is the number of charges that are responsible for shifting the Dirac voltage and CT is the gate capacitance. Now let us compare the flat and deformed graphene. The data in Fig. 4e suggest that the EDL capacitance of the deformed graphene is ~2.4 times smaller than the flat graphene. The data in Fig. 2e shows a Dirac voltage shift of ~80 mV for the flat graphene and a shift of ~180 mV for the deformed graphene when measuring 2 μM DNA. The ratio of the number of charges (that cause the Dirac voltage shift) of the flat (F) to deformed (DE) graphene sensors is given by $Dn_F/Dn_{DE} = (CT_{F,DVD,F}) / (CT_{DE,DVD,DE}) = 80 \text{ mV} \cdot 2.4 / 180 \text{ mV} = 1.07$. Based on this simple calculation, the total number of charges that contributed to the shift of the Dirac voltage are almost the same for the flat and deformed graphene sensors in this example. This is inconsistent with the hypothesis that the enhanced sensitivity is due to the increase of Debye length and hence the increase of the number of charges.

Response: As the Reviewer correctly pointed out, the ratio of Dirac point shift of the crumpled graphene to that of the flat graphene is $180\text{mV}/80\text{mV}=2.25$ which leads to almost identical charge carrier density change for the highest concentration of DNA (2 μM). However, this is only true for very high DNA concentrations. We think the reason for this is that at high concentrations (e.g. 2 μM), the adsorbed charges are saturated on the surface of both flat and crumpled graphene giving rise to identical charge transfer. Therefore, addition of charges is simply accumulated in layers away from graphene without being transferred to the graphene. To investigate this adsorption saturation, we

plotted the area per nucleotide (packing) for the concentrations used in this study in Figure S1 assuming all DNA molecules formed a monolayer at the graphene interface. As shown, for concentrations higher than ~ 1 nM, the area per nucleotide is lower than the area occupied by a DNA nucleotide (based on the size of one nucleotide) indicating an extreme packing if the molecules were hypothetically adsorbed as a monolayer next to graphene. Therefore, DNA adsorption saturation must take place for high concentrations (~ 1 nM and above) and the additional molecules are simply accumulated in layers away from graphene.

We further investigated the Dirac point shift for a range of different concentrations to identify the dominant mechanisms by which the shift takes place. First, we excluded the effect of the band gap and calculated the shift solely based on the charge transfer from the unscreened DNA molecules using $\Delta V_D = \frac{eN_{DNA}^{unscreened}}{C_T}$. By matching the experimental ΔV_D , charge transfer ($N_{DNA}^{unscreened}$) can be extracted. The ratio of the transferred charge to the total available DNA charge in the solution is plotted for different concentrations in Figure 2a. As shown, for the crumpled graphene, the charge transfer is higher than that of a flat graphene (indicating less screening in crumpled graphene) for the intermediate range of concentrations. In addition, the required charge transfer exceeds the maximum available charge ($>100\%$) for low concentrations indicating that charge transfer by its own is not the only mechanism responsible for the change in the carrier charge density. Next, we assume a constant change in the charge carrier density corresponding to a Dirac point shift of -0.042 V due to the band gap change (we justify the band gap opening in the response to the next comment). In Figure 2b, with the band gap included, the % of the charge transfer is estimated by matching the experimental shifts. As shown, for the crumpled graphene, for the two highest concentrations, the charge transfer is almost zero as the band gap is the dominant contributor to the shift. In Figure 2c, we plotted individual contribution of charge transfer and bandgap (note that due to the complexity of band gap opening, we assume a constant Dirac point shift due to band gap opening). In Figure 2d, the experimental data is replotted where we divided the concentration into three regions. < 200 aM region where the band gap opening is dominant, >200 aM and <1 nM region where the charge transfer becomes significant in addition to the band gap and >1 nM region where the charge transfer is dominant while DNA adsorption saturation on graphene takes place.

Following our discussion on the adsorption saturation, we can postulate the following scenario;

If we assume the following;

1. Spatial ratio of concave:convex is 1:1.
2. Concave is the origin of sensitivity enhancement. Convex is similar to flat.
3. DNA prefers to land on concave as the energies between DNA and graphene are -532.187 kcal/mol at concave and -467.484 kcal/mol at convex. This is from our simulation result (Fig. 3).

So, when we add DNA from low to high concentration, possibly DNA will first land on concave region at low concentrations (because of the lower energy). As we keep adding DNA with higher concentration, the concave region could become saturated first, and then the DNA would settle on convex regions. The screening effect on convex region is similar to

that on flat surfaces, so it does not contribute to sensitivity enhancement (or signal size amplification). Therefore, after a certain concentration, we cannot see the enhancement and the slope become similar with the flat surface.

It is hard to tell the exact point of saturation of the concave region but we are sure (based on our simulation results) that the improvement of the device performance only happens when DNA molecules fill the concave regions. After the available concave regions has been fully, there is no more enhancement.

Also, it is possible that part of the sensor area is not accessible to DNA due to physical constrictions created by the crumpling process, hence reducing the effective area of the crumpled graphene. If that was the case, the total number of DNA on the graphene can be somewhat different and it can affect the signal size.

We believe that our experimental data reflects multiple complex effects coming into play due to the crumpling and extreme curvatures. These include modulation of Debye screening, changes in dissociation constant, bandgap opening, etc.

We included Figure S1 and Figure S2 to the supporting information and the following sentences to the Supplementary Note on DNA adsorption to the graphene surface: “We investigated the DNA adsorption onto the graphene by plotting the area per nucleotide (packing) for the concentrations used in this study in Figure S1 assuming all DNA molecules formed a monolayer at the graphene interface. As shown, for the concentrations higher than 1 nM, the area per nucleotide is lower than the area occupied by a DNA nucleotide (based on the size of one nucleotide) indicating an extreme packing if the molecules were hypothetically adsorbed as a monolayer next to graphene. Therefore, DNA adsorption saturation must take place for high concentrations (~1nM and above) and the additional molecules are simply accumulated in layers away from graphene.

We studied the Dirac point shift for a range of different concentrations to identify the dominant mechanisms by which the shift takes place. First, we excluded the effect of the band gap and calculated the shift solely based on the charge transfer from the unscreened DNA molecules using $\Delta V_D = \frac{eN_{DNA}^{unscreened}}{C_T}$. By matching the experimental ΔV_D , charge transfer ($N_{DNA}^{unscreened}$) can be extracted. The ratio of the transferred charge to the total available DNA charge in the solution is plotted for different concentrations in Figure 2a. As shown, for the crumpled graphene, the charge transfer is higher than that of a flat graphene (indicating less screening in crumpled graphene) for the intermediate range of concentrations. In addition, the required charge transfer exceeds the maximum available charge (>100%) for low concentrations indicating that charge transfer by its own is not the only mechanism responsible for the change in the carrier charge density. Next, we assume a constant change in the charge carrier density corresponding to a Dirac point shift of -0.042 V due to the band gap change (we justify the band gap opening in the next section of this supplementary information). In Figure 2b, with the band gap included, the % of charge transfer is estimated by matching the experimental shifts. As shown, for the crumpled graphene, for the two highest concentrations, the charge transfer is almost zero as the band gap is the dominant contributor to the shift. In Figure 2c, we plotted the individual contribution of charge transfer and bandgap (note that due to the complexity of

band gap opening, we assume a constant Dirac point shift due to band gap opening). In Figure 2d, the experimental data is replotted where we divided the concentrations into three regions. < 200 aM region where band gap opening is dominant, >200 aM and <1nM region where the charge transfer becomes significant in addition to the band gap and >1nM region where the charge transfer is dominant while DNA adsorption saturation on graphene takes place.

Figure S1 The packing of DNA molecules adsorbed onto the graphene surface. The area per nucleotide is plotted along the y-axis if the entire DNA concentration were to be adsorbed as a single layer on the graphene surface. The area occupied by one DNA nucleotide is assumed to be $\sim 8 \text{ nm}^2$ based on DNA nucleotide size (the dashed line). Concentrations higher than 10^{-9} M results in extreme packing (area per nucleotide smaller than nucleotide size) which indicates saturation of a monolayer of DNA molecules on the graphene surface.

Figure S2 Percentage of DNA charge transfer to graphene to match the experimental Dirac point shift without band gap contribution (a) and with band gap contribution (b). Without band gap, the required charge transfer exceeds the maximum available limit indicating that charge transfer is not the only mechanism by which Dirac point shift occurs. c) contribution of charge transfer and band gap (a constant Dirac shift due to band gap is assumed) to the total Dirac point shift. d) Three different regions are defined. < 200 aM region where the band gap opening is dominant, >200 aM and <1nM region where the charge transfer becomes significant in addition to the band gap opening and >1nM region where the charge transfer is dominant while DNA adsorption saturation on graphene takes place.

- The authors suggest that the increase of Debye length cannot alone explain the observed enhanced sensitivity. Therefore, they offer another possible factor that might be responsible for the increased sensitivity, which is the bandgap opening of graphene at the functionalization sites. To justify this, they started with the Boltzmann transport theory in the diffusive regime of graphene, where the mobility is inversely proportional to the density of charged impurity scatterers (n_{imp}). Then they somehow connect this phenomenon to another one where the mobility of graphene decreases with increasing its energy gap. Please explain how these two phenomena are related? Moreover, the argument of functionalization on the armchair graphene is ambiguous. Please further elaborate. If the authors want to go in this direction, it is easy to estimate the mobility and show its relationship with n . Then they must establish how local change in the bandgap of graphene influences the transfer characteristics I_d - V_g of the device and establish this on the measured characteristics of the sensors. Note that the shift of the Dirac voltage, which is used here as the measure of sensitivity, is a macroscopic effect and is given by $e \cdot Dn / CT$. Mobility does not appear in this relationship although it might be a function of Dn . However, it is unclear from the description of the manuscript how Dn (number of charges) in the equation for DVD is a function of mobility.

Response: As mentioned in the response to the previous comment, the shift without including any effect from the bandgap opening cannot explain the Dirac point shift for ultralow concentrations. There must be some other mechanism in addition to the charge transfer to induce changes in the charge carrier density of graphene. Band gap opening upon DNA addition is considered as the possible explanation. The change in the carrier charge density of graphene due to a bandgap change after adding DNA molecules can be obtained from $\Delta n_{\text{bandgap}} = \frac{1}{e\rho} \left(\frac{1}{\mu^{\text{DNA}}} - \frac{1}{\mu^{\text{no DNA}}} \right)$, where e is the elementary charge and ρ is the resistivity. Wang et al.¹ showed that electronic mobility (μ) in graphene decreases with increasing bandgap where mobility is obtained empirically from; $\mu = 0.114 \times 10^4 E_g^{-\frac{3}{2}}$ (E_g in eV and μ in $\text{cm}^2/\text{V-s}$). Since the bandgap changes are local, the global (macroscopic) change in the charge carrier density can be estimated by $\Delta n_{\text{global}} = b \Delta n_{\text{bandgap}}$, where b is the area fraction of the affected regions. Assuming a bandgap change from 0.4224 eV to 1.7641 eV in $10^{-7}\%$ of the crumpled graphene (each DNA nucleotide must at least affect an area of 39.9 nm^2 for 2aM concentration), the shift is noticeable and is estimated to be $\sim 10\text{mV}$ (using $\Delta V_D = \frac{e\Delta n_{\text{global}}}{C_T}$). $39.9 \text{ nm}^2/\text{nucleotide}$ (an area with a diameter of 7.1nm) is relatively large compared to the size of a single DNA nucleotide indicating a larger-range effect. To ensure the bandgap change is indeed long-range, we investigated the effect of graphene size on the bandgap opening due to a single DNA base. As shown in Figure S3, we first considered one unit cell (width of ~ 12 and length of ~ 12) and computed the bandgap of crumpled graphene in the presence of A-base (with the orientation 1 as indicated in table 1 of the manuscript). Then, keeping the single A-base, we increased the number of the crumpled graphene unit cells to 2, 3 and 4. For example, a 4-unit cell is a supercell with a width of ~ 12 and a length of ~ 48 . Note that the width of all the unit cells is ~ 12 and only the length is varied. Based on this study, we observe that the bandgap for the case of 4-unit cell is about 50% of that of a single unit cell which is still a significant bandgap value. This supports our hypothesis that the local change of bandgap is long-range and influences the electronic properties of graphene globally.

We included Figure S3 in the supporting information and added the following sentences to the Supplementary Note on Modeling of Dirac point shift for band gap opening: “The shift is directly obtained from $\Delta V_D = \frac{eN_{\text{DNA}}^{\text{unscreened}}}{C_T}$ without including any effect from bandgap. However, for the ultralow concentrations (e.g., 2 aM), the shifts are too small to be noticed without including the effects from bandgap opening (Supplementary Fig. 9c-d and Fig. 10a-d). The carrier charge density change of graphene due to a bandgap change after adding DNA molecules can be obtained from $\Delta n_{\text{bandgap}} = \frac{1}{e\rho} \left(\frac{1}{\mu^{\text{DNA}}} - \frac{1}{\mu^{\text{no DNA}}} \right)$, where e is the elementary charge, and ρ is the resistivity. Wang et al.¹ showed that electronic mobility (μ) in graphene decreases with increasing bandgap where mobility is obtained empirically from $\mu = 0.114 \times 10^4 E_g^{-\frac{3}{2}}$ (E_g in eV and μ in cm^2/Vs). Since the bandgap changes are local, the global (macroscopic) change in the charge carrier density is estimated by $\Delta n_{\text{global}} = b \Delta n_{\text{bandgap}}$, where b is the area fraction of the affected regions. For the 2 aM concentration, assuming a bandgap change from 0.4224 eV to 1.7641 eV in $10^{-7}\%$ of the crumpled

graphene where each DNA nucleotide affects at least an area of 39.9 nm^2 (see the calculation below), the shift is noticeable and is estimated to be $\sim 5 \text{ mV}$ (using $\Delta V_D = \frac{e\Delta n_{\text{global}}}{C_T}$) as shown in Supplementary Fig. 9e. $39.9 \text{ nm}^2/\text{nucleotide}$ (an area with a diameter of 7.1 nm) is relatively large compared to the size of a single DNA nucleotide. To ensure the bandgap change is indeed long-range, we investigated the effect of graphene size on the bandgap opening due to a single DNA base. As shown in Figure S3, we first considered one unit cell (width of ~ 12 and length of ~ 12) and computed the bandgap of crumpled graphene in the presence of A-base (with the orientation 1 as indicated in Table 1 of the manuscript). Then, keeping the single A-base, we increased the number of the crumpled graphene unit cells to 2, 3 and 4. For example, a 4-unit cell is a supercell with a width of ~ 12 and a length of ~ 48 . Note that the width of all the cells is ~ 12 and only the length is varied. Based on this study, we observe that the bandgap for the case of 4-unit cell is about 60% of that of a single unit cell which is still a significant bandgap value. This supports our hypothesis that the local change of bandgap is long-range and influences the electronic properties of graphene globally.”

Figure S3. Bandgap as a function of the size of graphene unit cell for a single DNA base. The bandgap is normalized by the bandgap of 1-unit cell. The width of all the cells is ~ 12 and only the length is varied. The graphene bandgap in 4-unit cell is about 60% of that of 1-unit cell which is still a significant bandgap value. This shows that the effect of a single DNA base on the bandgap of graphene is long range.

Overall, the observed enhanced sensitivity of deformed graphene is interesting. However, the discussion on the origin of this phenomenon requires further elaboration.

Response: Thank you for the comments. Hope our responses can address the reviewer’s concerns and that the paper would not be acceptable for publication.

Reviewer #3 (Remarks to the Author):

The paper by Hwang et al. presents an experimental and theoretical study of using crumpled graphene based FET to detect nucleic acids. The authors claimed using crumpled graphene can achieve good sensitivity in DNA detection and allow fast DNA sensing in millimeter scale. Although the discovery is interesting and could open new strategies for realizing the potential of graphene (and other two-dimensional materials) in biosensing applications, the paper can not be published in the current form. I think the paper can be considered to be published in Nature Communications if the authors can address the following concerns well.

Overall, the paper is well written with systematic experimental and theoretical investigations. However, a central question to this paper is: can this method achieve fast and reliable base-specific detection of DNA? This question is a key challenge in this field. Throughout the paper the authors claim to have found a ultrasensitive sensing method and conclude that crumpled graphene have higher sensitivity than other 2D material counterparts, but can the electric signals generated by this approach be base-specific (i.e. each nucleobase on graphene FETs leads to distinct electronic responses)? In the theoretical part (especially in Table I), I notice that the band gap opening on graphene is almost similar for all nucleobases. The authors should address this point to justify the novelty they present in this study.

Response: We appreciate the reviewer's comment but would like to respectfully point out that the point of the paper was not to directly detect specific bases for sequencing. We aim to develop a biosensor that can detect target molecules by binding with a probe that can capture the target. Of course, the capture of the capture molecules would be sequence specific. But we are not aiming to detect specific individual nucleotides with distinct electronic signatures for DNA sequencing per say. The reasons for showing the simulations for bandgap opening on graphene with specific nucleobases was that those bases do open the band-gap, which then creates electrical currents when the target DNA molecules are close to those bent surfaces and can create the local electrical hot-spots.

The quest for detection of individual bases is a different one and not the goal of this study.

We hope we have clarified this point to the reviewer's satisfaction.

I also have some other questions and some minor comments on the paper:

1. The morphology of crumpled graphene is quite complicated with various concave, convex and confined regions. My concern is how stable the structure is and whether the process can be reproduced conveniently. Can author explain how they determine the structure is still crumpled "monolayer" graphene after the process?

Response: Thank you for this question. The crumpling method itself that we used in this manuscript has been published earlier (Nano Lett. 2015 15, 3, 1829-1835, Nano Lett. 2015 15, 11, 7684-7690). It is a robust and reliable process and does not require any complicated lithography or sophisticated protocols. The SEM images in the manuscript show that structure of the crumples are similar to what has been reported in previous publications.

Moreover, the resistance levels are similar ($\sim 10 \text{ K}\Omega$) over the devices used in all experiments and all the devices show the same sensitivity in detection of biomolecules.

As shown in the Fig. 1d and Supplementary Fig. 3a-b, we performed Raman Spectroscopy to confirm the structure of the graphene layer. However, as polystyrene (PS) substrate has a Raman peak close to graphene G-peak, the Raman peak marked as “G” in the plot is actually ‘graphene G peak + PS’. To observe the G peak without the peak from PS, we prepared flat and crumpled samples on VHB substrate (because VHB does not have Raman peak close to graphene G peak). The following is the sample preparation protocol;

1. Transfer graphene on a PDMS stamp
2. Transfer graphene onto VHB substrate.
 - i. For Flat sample: (VHB tape was put on a slide glass because the soft VHB nature made hard the contact printing process.)
 - ii. For crumpled sample: 100% prestrain was applied in x- and y-axes to apply same amount of prestrain as crumpled graphene on PS substrate.

The result graphs are below;

Based on 2D/G intensity ratio (d), both crumpled and flat samples showed larger than 2. FWHM of 2D peak was analyzed, and in both samples, FWHM of 2D peak indicates the presence of monolayer graphene. (For monolayer graphene, FWHM < 30cm⁻¹, Nano Lett. 2007, 7, 2, 238-242). VHB has a Raman peak at ~2670 cm⁻¹ on crumpled sample, and it overlaps with graphene's 2D peak. That would have made crumpled graphene sample's FWHM value larger than flat sample. 2D peak center was also analyzed to be larger from crumpled sample (2636.4 cm⁻¹ vs. 2628.2 cm⁻¹) for similar reasons. Based on our Raman spectroscopy analysis, both crumpled and flat graphene samples were monolayer. In addition, since crumpled graphene is a three-dimensional structure, out-of-plane structures may interact or even touch each other, however, the spectral analysis does not indicate apparent evidence of local folding/touching/ interacting of graphene structures. We have added this result in the supplementary Note. Also, there is a previous report that the

graphene remained monolayer after the same crumpling process (Adv. Mater. 2016, 28: 4639-4645). We hope this addresses the concerns raised by the reviewer.

2. The "electric hotspots" resulting from deformed graphene structure is expected. This could also be done with other confined setups such as graphene nanopores. The author could review these methods and address why using crumpled graphene is better.

Response: While we agree that other methods such as graphene nanopores could also create surfaces that have high curvatures, those high curvatures would be around the edges of the nanopores in a suspended membranes and hence require an entirely different device architecture, fabrication process, and for a different application, which is DNA sequencing. Our 'hot spots' are electrical and on the continuous film and we believe are created not only by the formation of the valleys and trenches in the crumpled and wrinkled graphene, but are also formed due to EDL changes, and the band gap opening. Moreover, we believe the crumpled graphene structure is entirely different than the nanopores as these crumpled films are on a surface and not suspended with pores drilled through them. Our process has several benefits over other confined setups such as nanoribbon or nanopore in terms of fabrication. First, it does not require complicated lithography and E-beam process, which are required to fabricate the other nano-confined devices mentioned by the reviewer. The fabrication process of the crumpled graphene is processed by macroscopic manipulation of the materials and the device does not need to be 'nano'-sized to enhance its sensing performance, which allows facile fabrication and reliable reproducibility. We have highlighted this in the manuscript and added references of other structures such as nanopores. We would also like to point out that our groups has extensive experience in the fabrication and modeling of nanopore structures for DNA sequencing and the process and structure reported here in this manuscript is entirely different in the fabrication and the application.

3. The authors present the comparison among different adsorption scenarios of nucleobases on graphene in Figure 3. However, the presentation in (e-h) is confusing. The authors state that concave adsorption leads to less electrical screening of DNA molecules due to strong contact with underlying graphene. How is this statement reflected in Figure 3 (e) - (h)? It seems to me the screening factor curve that the author used to quantify the sensitivity for all cases in Figure 3(e) - (h) is similar to each other. I cannot conclude that the concave surface (Figure 3(f)) results in superior sensitivity than other cases. Authors need to clarify this and revise the graphical presentation of their MD results.

Response: The screening factor curve looks similar for all the cases as expected. However, the key point in Figure 3 (e)-(h) is the relative position of DNA charges with respect to ions. The screening by ions starts at a larger distance away from the graphene surface in the concave case leaving much of DNA charges next to the surface unscreened. In the concave region, DNA molecules are adsorbed to the surface (with the highest adsorption energy, Figure 9 of the supplementary) excluding ions from the surface due to the strong concave confinement. In the other cases (flat, convex and across), however, most of DNA charges are located further away where ionic screening take place.

For further clarification, we modified the following sentences in the manuscript on page 6:

“...As shown in Fig. 3a-h, because of the confined nature of the concave region, ions are excluded and are farther away from the concave graphene surface leading to increased exposure of the DNA to the graphene surface. Here, the relative position of DNA charges with respect to the ions matters. In other words, the screening by ions starts at a larger distance away from the graphene surface in the concave case leaving much of DNA charges next to the surface unscreened. This results in weaker ionic screening of DNA molecules...”

4. The authors only consider single stranded DNA in MD. What would be the response on double stranded DNA? double stranded DNA could lead to significant difference compared with single stranded DNA (see Kabelac et al. Phys. Chem. Chem. Phys. 2012, 14, 4217-4229).

Response: We considered single-stranded DNA as the Dirac point shift corresponds to the addition of complementary single-stranded DNA to the solution. In the MD simulations, we studied how different surface morphologies (e.g., flat, concave and convex) affect the adsorption of DNA molecules on graphene. As shown in Figure 3 of manuscript and Figure 9 of the supplementary, the adsorption of single-stranded DNA molecules is strongest in the case of the concave graphene surface. Regardless of whether a single-stranded or a double-stranded DNA is used, the adsorption should be higher in the concave region as the type (or chemistry) of molecule is the same and the only varying condition is the morphology of the surface. We, however, note that the degree of adsorption might be different for double-stranded DNA molecules as suggested by the Reviewer. We should also note that due to the complexity of actual graphene structure in experiments, it is impossible in MD simulation to calculate the exact value of screening or degree of adsorption based on the simulations for one type of crumpling (considering the wavelengths and amplitudes used in the simulations). Therefore, we are only interested in the relative degree of DNA adsorption on different graphene surfaces.

For further clarification, we modified the following sentences in the manuscript on page 6: “...The interaction energies show that the adsorption of DNA to graphene in the concave region is the strongest. The calculated energies for the concave, convex, and across cases are -532.187 kcal/mol, -467.484 kcal/mol, and -416.308 kcal/mol, respectively (Supplementary Fig. 11). We should note that due to the complexity of actual graphene structure in experiments, it is difficult in MD simulations to calculate the exact degree of adsorption based on the simulations for a single type of crumpling (the graphene surfaces considered in the MD simulations). Therefore, we only investigated the relative degree of adsorption on different graphene surfaces for a single-stranded DNA molecule. See Supplementary Fig. 12 for more detail about DNA adsorption onto graphene surface.” (similar observations should be expected for double-stranded DNA molecules as the nature of interaction is similar, even though double stranded DNA has less π - π stacking interaction with graphene).

5. The authors list the comparison between the band gap of flat and crumpled graphene upon adsorption of nucleobases in different orientations. However, what is the exact orientation of nucleobases on graphene considered in DFT calculations? Authors should provide some explanations on this.

Responses: We thank the Reviewer for this question. In orientation 1 and orientation 2, the plane of the base ring is parallel and perpendicular to the graphene sheet, respectively. The plane of the base ring in orientations 3 and 4 is between the horizontal and vertical orientations (orientation 1 and 2). We have provided snapshots of orientation 1 and 2 in Figure S4.

We included Figure S4 in the supporting information and the following sentences in the Supplementary Note on DNA orientations in Aab Iinito calculations: “In orientation 1 and orientation 2, the plane of the base ring is parallel and perpendicular to the graphene sheet, re

Figure S4. The nucleobase orientation above the graphene surface used in the DFT/GW simulations. a) orientation 1 where the plane of the base ring is parallel to the graphene sheet and b) orientation 2 where the plane of the base ring is perpendicular to the graphene sheet.

6. The electronic signals of graphene can be significantly altered by changing the local contact between graphene and nucleobases (see Ahmed et al. Nano Lett., 2012, 12, 927-931, Yin et al., J. Phys. Chem. Lett. 2017, 8, 3087-3094 and Caridad et al., Carbon, 2018, 129, 321-334). When considering modelling the actual contact between DNA nucleobases and flat graphene/crumpled graphene, how did authors determine the three representative cases? The authors need to provide more information of how the graphene/nucleobase system is modelled. Moreover, is the nucleobase weakly adsorbed on graphene via vdW forces? The authors do not mention whether they have use vdW corrections in DFT calculations.

Response: The structure of the flat/crumpled graphene with a nucleobase is relaxed (optimized) using DFT with LDA exchange correlations. The forces between the atoms are minimized to 0.01 eV/Å with a total energy convergence tolerance of 10^{-6} eV. The details of the optimization method are included in the DFT/GW method section of the main manuscript. Four different orientations of nucleobases above the surface of graphene are considered to ensure different possible contact interfaces are covered in our study.

We did not include any vdW corrections in the DFT simulations. The vdW corrections + DFT is a possible method to improve the bandgap calculations estimated by DFT. However, to compute bandgap with a high accuracy, GW simulations, which are based on many-body interactions and are superior to DFT, were performed (see Table 1 of the manuscript).

7. Another concern is the missing of data in Table I. GW results are only given for some cases.

Response: We thank the Reviewer for highlighting this. The data point of flat graphene with a T base is now included in the revised manuscript. Since GW is computationally expensive, we performed GW only for orientation 1 on all graphene surfaces as well orientation 2 on crumpled armchair graphene.

8. Among available data in Table I, I notice a great difference between band gap of crumpled graphene upon adsorption of Adenine in Orientation I and II (1.21 eV) via GW, while the band gap difference is only less than 1 meV with DFT-LDA? I wonder if this result is reasonable. From the authors' results, the GW results seems to only increase the band gap magnitude for all cases, not resulting in big band gap differences between different orientations. Authors need to clarify this.

Response: The Reviewer has brought an important point that we should highlight in the main manuscript. In orientation 1, the plane of the nucleobase ring is parallel to the graphene maximizing the DNA-graphene interfacial area. However, in orientation 2, the plane of the base ring is perpendicular to the graphene surface minimizing the interfacial area as shown in Figure S5. The nucleobase above the crumpled graphene induces a dipole due to the charge transfer to the graphene. The parallel configuration (orientation 1) is expected to have a stronger induced dipole compared to the perpendicular configuration (orientation 2) due to larger interfacial area with more nearest neighbor interactions. Hence, a wider bandgap is expected for orientation 1. The dependence of induced dipole on geometry is discussed by Park et al.².

GW, based on many-body interactions, is superior to the DFT, which is based on a simpler single electron formulation. The lack of the electron-electron interactions in DFT-LDA is responsible for similar bandgap values obtained for the different configurations in Table 1 of the manuscript. However, distinct bandgaps are obtained using GW. DFT solves Kohn-Sham equation that is given by

$$[T + v(x)]\Psi_n = \varepsilon_n\Psi_n$$

where T is the kinetic energy operator, $v(x)$ is the external energy potential, Ψ_n is the wavefunction of n eigenstate, and ε_n is the eigenenergy level of state n . The Kohn-Sham approximation is based on non-interacting single electron. On the other hand, GW considers the many-body interaction as follows

$$[T + v(x) + v_H]\Psi_{nk}(r) + \int d^3r' \Sigma(r, r', E_{nk}) \Psi_{nk}(r') = E_{nk}\Psi_{nk}(r)$$

where v_H is Hartree potential and E_{nk} is quasiparticle energy level that is associated with orbital Ψ_{nk} . The main difference between DFT and GW is the many-body self-energy operator Σ from Green's function, polarizability and screened Coulomb interaction function. This operator allows GW to account for the electron-electron interactions resulting in accurate exchange energies. Since the parallel and perpendicular configurations (shown in Figure S5) have different Coulomb screening and different

induced dipole, GW can capture the difference in bandgap arising from a different configuration.

The following sentences are now included in the Supplementary Note on DNA orientations in *Ab initio* calculations: “Since GW is computationally expensive, we performed GW only for orientation 1 on all graphene surfaces as well as orientation 2 on crumpled armchair graphene. Comparing the DFT and GW bandgaps of crumpled graphene, we note that the DFT bandgaps for orientations 1 and 2 are almost identical. In orientation 1, the plane of the nucleobase ring is parallel to graphene maximizing the DNA-graphene interfacial area. However, in orientation 2, the plane of the base ring is perpendicular to the graphene surface minimizing the interfacial area as shown in Figure S5. The nucleobase above the crumpled graphene induces a dipole due to the charge transfer to graphene. The parallel configuration (orientation 1) is expected to have a stronger induced dipole compared to the perpendicular configuration (orientation 2) due to the larger interfacial area with more nearest neighbor interactions. Hence, a wider bandgap is expected for orientation 1. The dependence of induced dipole on geometry is discussed by Park et al.²”

Figure S5. The nucleobase orientations above the crumpled graphene surface used in the DFT/GW simulations. a) orientation 1 where the plane of the base ring is parallel to the a-c plane and b) orientation 2 where the plane of the base ring is perpendicular to the a-c plane (or parallel to the b-c plane).

Minor comments:

9. Figure 1(e): Is this the charge transfer characteristic of crumpled graphene FET (same to Supplementary Figure 2(f))? Authors should make this clear in the main text and Figure caption.

Response: Thank you for pointing this out. Yes Fig. 1e and Supplementary Fig. 2f are same graph and it is the crumpled graphene FET. We have added the clarification on the caption of Fig. 1.

10. When discussing the graphene-based DNA sensing and sequencing, the authors can use this well-written review: Heerema et al., Nature Nanotech., 2016, 11, 127-136.

Response: Thank you for the valuable suggestion. Now we have added the paper in the reference list.

Reviewers' comments:

Reviewer #1 (Remarks to the Author):

My concerns and comments were fully addressed in this revised version. I do also appreciate the details and the precise discussion of all reviewers concerns in the response to reviewers letter.

To my opinion the manuscript can be accepted and published as a very interesting piece of work in Nature Communications.

Reviewer #2 (Remarks to the Author):

The authors have address my major comments and questions and included the relevant discussions.

Reviewer #3 (Remarks to the Author):

Thanks for the authors' responses and revisions on the manuscript. I appreciate that most of the concerns and technical comments have been addressed. However, I still have some questions on the current form of the manuscript.

1. On Figure 3 and related texts: although authors have added explanations on why concave contact is different from other cases. it would still be difficult for readers to understand from the current graphic presentations in Figure 3. I suggest the authors do the following to improve Figure 3:

- Rearrange the subfigures (a-h) in this order: (a-d) on the first column and (e-g) on the second column. In this way, the screen factor curve would be next to its corresponding schematic.
- Highlighting the distance where the ion screening starts in the figure (e-g) to reflect the explanations in the main texts.
- Reorganize the caption of Figure 3 to make it concise and clear.

2. On DFT part, I have these comments:

- Table I should be removed from the main texts. The purpose of Table I is to show that the crumpled graphene can open significant band gap compared to flat graphene. Considering most of the information in the table can only be well understood by reading the Supplementary Notes and Figures, I suggest to move this table to Supplementary Information.
- Figure S17 and S18: for parallel orientation, the author mentioned in the Supplementary Note: "...due to the larger interfacial area with more nearest neighbor interactions". It seems to me that the distance between neighboring molecules is too far to have considerable neighboring interactions? (if the simulation box is 12*12 Angstrom, the nearest distance between the nucleobases should be about 9 Angstrom. At this distance, the intermolecular distance should be negligible.) Authors should clarify on this.
- Overall, I am not satisfied with the Supplementary Note. The two points author want to make: a) crumpled graphene have larger band gaps (also key message for the main texts) b) orientation of nucleobases can play a role in affecting the band gap. Authors should spend a bit more texts explaining on how dipoles relates to configuration and band gap (not just mentioning some references, the author may also look for other papers on this topic). In addition, the author can include a schematic of the Orientation 3 and 4 and include the data on these orientations.

Authors should also check the main texts and Supplementary Information very carefully for minor mistakes and formatting errors.

Response to the reviewer's comments

Comment: 1. On Figure 3 and related texts: although authors have added explanations on why concave contact is different from other cases, it would still be difficult for readers to understand from the current graphic presentations in Figure 3. I suggest the authors do the following to improve Figure 3:

- Rearrange the subfigures (a-h) in this order: (a-d) on the first column and (e-g) on the second column. In this way, the screen factor curve would be next to its corresponding schematic.
- Highlighting the distance where the ion screening starts in the figure (e-g) to reflect the explanations in the main texts.
- Reorganize the caption of Figure 3 to make it concise and clear.

Response: We thank the reviewer for these suggestions. We rearranged the subfigures and highlighted the point where the screening starts to take place with an arrow as shown in Figure 3 of the main manuscript (see also below for Figure 3). We also added a sentence describing the arrow drawn in the figures.

Figure 3. The schematic of the simulations for equilibrated DNA on **a**, flat graphene **b**, concave surface of crumpled graphene **c**, convex surface of crumpled graphene and **d**, across the graphene crumples. Graphene is shown in blue, ions are presented as cyan and yellow spheres and the DNA bases are shown in different colors. Water molecules are not shown for better presentation. The molar concentration of

ions (sodium and chloride) and the backbone of DNA strand along with the screening factor of ions are plotted as a function of the distance from the graphene surface for **e**, flat **f**, concave **g**, convex and **h**, across configurations of DNA. The location where the ionic screening starts to take place is shown using an arrow. In the concave region, ions are excluded due to its confinement and most of the adsorbed DNA molecule remains unscreened electrostatically. Less screening increases $N_{\text{DNA}}^{\text{unscreened}}$ and induces more charge density in graphene resulting in a larger Dirac point shift. **i**, The 2.45nm-diameter CNT that is used to model a narrow trench in crumpled graphene is shown with CNT and graphene carbon atoms in blue, ions in cyan and yellow, water molecules in red and DNA strand bases in different colors. The DNA adsorbs to the bottom of the trench and excludes ions near the surface (maximizing $N_{\text{DNA}}^{\text{unscreened}}$). The resulting giant electric potential modifies the carrier charge density of graphene. The potential is obtained from $V(z) = - \iint_{z_0}^z \frac{q(z)}{A \epsilon_0} dz dz$, where $q(z)$, A and ϵ_0 are the net charge of the system (ions, DNA and water) in z , surface area of the bottom of the trench and vacuum dielectric constant, respectively.

Comment: 2. On DFT part, I have these comments:

- Table I should be removed from the main texts. The purpose of Table I is to show that the crumpled graphene can open significant band gap compared to flat graphene. Considering most of the information in the table can only be well understood by reading the Supplementary Notes and Figures, I suggest to move this table to Supplementary Information.

Response: We appreciate the reviewer's insight. We moved Table I to the supplementary information.

Comment: - Figure S17 and S18: for parallel orientation, the author mentioned in the Supplementary Note: "...due to the larger interfacial area with more nearest neighbor interactions". It seems to me that the distance between neighboring molecules is too far to have considerable neighboring interactions? (if the simulation box is 12* 12 Angstrom, the nearest distance between the nucleobases should be about 9 Angstrom. At this distance, the intermolecular distance should be negligible.) Authors should clarify on this.

Response: We thank the reviewer for raising this important point. Here, we clarify what we mean by large interfacial area with more neighbor interactions. We added Supplementary Figure 19 to illustrate the charge density difference at the interface for both the parallel and perpendicular orientations. We have added the following paragraph to the supplementary information file.

"The charge density difference of the system (i.e., graphene and nucleobase system), $\Delta\rho_s$, is defined as

$$\Delta\rho_s = \rho_s - \rho_g - \rho_n$$

where ρ_s is the charge density of the full system including both the crumpled graphene and the nucleobase in the unit cell, ρ_g is the charge density obtained by simulating the crumpled graphene without the nucleobase, and ρ_n is the charge density obtained by simulating the nucleobase without the crumpled graphene. $\Delta\rho_s$ represents the interfacial charge density due to the adsorption of the nucleobase onto the crumpled graphene. As shown in Supplementary

Figure 19, when the nucleobase is placed parallel to the crumpled graphene, the interfacial charge density is distributed along the nucleobase atoms. However, when the nucleobase is perpendicular to the crumpled graphene, the charge density is predominantly on the lower edge of the nucleobase experiences. Therefore, parallel orientation has a larger interfacial area with more Coulombic interactions (all atoms in the nucleobase directly interact with the crumpled graphene) compared to the perpendicular orientation (only the atoms towards the lower edge of the nucleobase have a direct interaction with the graphene surface). The different charge distributions and stacking lead to different electronic interactions for each orientation^{13,14}. Further, Supplementary Fig. 19 shows the local electrostatic and local Hartree potential of each orientation in the non-periodic direction (c-direction shown in Supplementary Fig. 19a). At the interface, for orientation 1, there is a high energy barrier compared to orientation 2, which results in a higher bandgap for orientation 1. Different interactions in these orientations lead to different band gaps as captured by the GW method which accounts for Coulombic interactions using a many-body approach^{15,16}.

Supplementary Figure 19. Interaction of the nucleobase with the crumpled graphene surface for different orientations: a) interfacial charge density for parallel orientation (orientation 1) and b) interfacial charge density for perpendicular orientation (orientation 2) (green and red colors represent negative and positive charge difference, respectively). Local potential versus the distance from the bottom of the crumpled graphene, c) for orientation 1 and d) for orientation 2 along the c -direction (red curve is the total electrostatic potential and blue curve is the Hartree contribution).

Comment: Overall, I am not satisfied with the Supplementary Note. The two points author want to make: a) crumpled graphene have larger band gaps (also key message for the main texts) b) orientation of nucleobases can play an role in affecting the band gap. Authors should spend a bit more texts explaining on how dipoles relates to configuration and band gap (not just mentioning

some references, the author may also look for other papers on this topic). In addition, the author can include a schematic of the Orientation 3 and 4 and include the data on these orientations.

Response: We thank the reviewer for bringing up these points which are very helpful to enhance the manuscript. We discuss each point separately below.

a) “Crumpled graphene have larger band gaps”: We have added the following paragraph and the Supplementary Figure 20 to the Supplementary information file.

“Crumpling graphene by introducing 1D periodic ripples is found to produce a bandgap opening¹⁻³. The opening of bandgap is attributed to the change in graphene curvature introducing quantum confinement with distinct electronic structures compared to the pristine/flat graphene³. In pristine graphene, the C-C bond length is ~ 1.41 Å for all carbon atoms with an angle of 120° which results in sp^2 hybridization. When graphene is crumpled, the bond length and the angles vary across the graphene. The optimized structure that we obtained using DFT shows that the C-C bond length has a value of 1.41 - 1.55 Å depending on the local curvature. In addition, the angles between the carbon atoms is found to be either 120° or 110° . The bond length of 1.55 Å and angle of 110° resemble the structure for sp^3 hybridized C atoms⁴⁻⁵. Thus, crumpled graphene contains sp^3 and sp^2 hybridization between C atoms. This is expected since the C atoms are not in the same plane due to crumpling. Further, the partial density of states of the pristine graphene (see Supplementary Figure 20) shows the p_x and p_z orbitals having sp^2 hybridization. By analyzing the partial density of states of the crumpled graphene (see Supplementary Figure 20), we observe overlapping between all three p orbitals (i.e., p_x , p_y and p_z) and between only two of them at a given energy level supporting the presence of both sp^3 and sp^2 hybridizations. Introducing sp^3 bonds produces strongly localized hybridization⁶⁻⁷ resulting in a bandgap opening. It should be noted that changing the degree of hybridization (i.e., electronic orbitals overlap) via changing the bonding interactions (e.g. bond length and angle) is found to play a significant role in controlling the conduction and valence energy band levels which leads to altered bandgaps in different materials⁸⁻¹².”

Supplementary Figure 20. Partial density of states for: a) pristine graphene and b) crumpled graphene for different molecular orbitals. Total density of states (gray filled), s-orbital contribution (red), px-orbital contribution (blue), py-orbital contribution (green), and pz-orbital contribution (orange) are shown.

b) “orientation of nucleobases can play a role in affecting the band gap”: We have revised the paragraphs which discussed the bandgap dependence on the nucleobase orientation. We also included the schematic of orientation 3 by revising Supplementary Figures 17 and 18. It should be noted that we do not have orientation 4 (it was a typo in our previous response) in Table 1 of the supplementary information file.

In orientation 1 and orientation 2 (see Supplementary Fig. 17 and Fig. 18), the plane of the base ring is parallel and perpendicular to the graphene surface, respectively. The plane of the base ring in orientation 3 is the same as in orientation 2. However, the nucleobase is rotated 180° as shown in Supplementary Fig. 17 and Fig. 18.

Since GW is computationally expensive, we only performed GW for orientation 1 on all the graphene surfaces and orientation 2 on the crumpled armchair graphene. Comparing the DFT and GW bandgaps of crumpled graphene, we note that the DFT bandgaps for orientations 1 and 2 are almost identical. However, the GW bandgaps are different for different orientations. To understand the interactions of the different orientations, we computed the interfacial charge density, $\Delta\rho_s$ (see the previous section of this response for the definition). Supplementary Fig. 19 shows the computed interfacial charge density for orientation 1 and 2. When the nucleobase is placed parallel to the crumpled graphene, the interfacial charge density is distributed across the nucleobase atoms. However, when the nucleobase is perpendicular to the crumpled graphene only the lower half of the nucleobase experiences the interfacial charge density and the upper half is screened. Therefore, orientation 1 has a larger interfacial area of Columbic interactions. The different charge distributions and stacking lead to different electronic interactions for each orientation^{13,14}. Supplementary Fig. 19 shows the local electrostatic and local Hartree potential of each orientation in the non-periodic direction (*c*-direction shown in Supplementary Fig. 19a). At the interface, for orientation 1, there is a high energy barrier compared to orientation 2, which results in a higher bandgap for orientation 1. The bandgap difference is captured by GW since it employs more accurate electrostatic calculations based on many-body interactions^{15,16}.

Supplementary Figure 17. The nucleobase orientations above the pristine graphene surface used in the DFT/GW simulations. a) orientation 1, where the plane of the base ring is parallel to the a-b plane, b) orientation 2, where the plane of the base ring is perpendicular to the a-b plane (or parallel to the a-c plane), c) orientation 3 is similar to orientation 2 where the ring is rotated 180° around the a-axis.

Supplementary Figure 18. The nucleobase orientations above the crumpled graphene surface used in the DFT/GW simulations. a) orientation 1, where the plane of the base ring is parallel to the a-b plane, b) orientation 2, where the plane of the base ring is perpendicular to the a-b plane (or parallel to the a-c plane), c) orientation 3 is similar to orientation 2 where the ring is rotated 360° around the a-axis.

Comment: Authors should also check the main texts and Supplementary Information very carefully for minor mistakes and formatting errors.

We have read the manuscript and supplementary information carefully and corrected the minor issues. We do appreciate such a detailed review.

References

1. Bai KK, Zhou Y, Zheng H, Meng L, Peng H, Liu Z, Nie JC, He L. Creating one-dimensional nanoscale periodic ripples in a continuous mosaic graphene monolayer. *Physical review letters*. 2014 Aug 18;113(8):086102.
2. Lee JK, Yamazaki S, Yun H, Park J, Kennedy GP, Kim GT, Pietzsch O, Wiesendanger R, Lee S, Hong S, Dettlaff-Weglikowska U. Modification of electrical properties of graphene by substrate-induced nanomodulation. *Nano letters*. 2013 Jul 19;13(8):3494-500.
3. Lim H, Jung J, Ruoff RS, Kim Y. Structurally driven one-dimensional electron confinement in sub-5-nm graphene nanowrinkles. *Nature communications*. 2015 Oct 23;6:8601.
4. Sofo JO, Chaudhari AS, Barber GD. Graphane: A two-dimensional hydrocarbon. *Physical Review B*. 2007 Apr 10;75(15):153401.
5. Boukhvalov DW, Katsnelson MI, Lichtenstein AI. Hydrogen on graphene: Electronic structure, total energy, structural distortions and magnetism from first-principles calculations. *Physical Review B*. 2008 Jan 22;77(3):035427.
6. Abrasonis G, Gago R, Vinnichenko M, Kreissig U, Kolitsch A, Möller W. Sixfold ring clustering in sp²-dominated carbon and carbon nitride thin films: A Raman spectroscopy study. *Physical Review B*. 2006 Mar 24;73(12):125427.
7. Zhang Y, Shang J, Fu W, Zeng L, Tang T, Cai Y. A sp²+sp³ hybridized carbon allotrope transformed from AB stacking graphyne and THD-graphene. *AIP Advances*. 2018 Jan 30;8(1):015028.
8. Miglio A, Heinrich CP, Tremel W, Hautier G, Zeier WG. Local bonding influence on the band edge and band gap formation in quaternary chalcopyrites. *Advanced Science*. 2017 Sep;4(9):1700080.
9. Jaffe JE, Zunger A. Theory of the band-gap anomaly in AB₂C₂ chalcopyrite semiconductors. *Physical Review B*. 1984 Feb 15;29(4):1882.
10. Khoa DQ, Nguyen CV, Bui LM, Phuc HV, Hoi BD, Hieu NV, Nha VQ, Huynh N, Nhan LC, Hieu NN. Opening a band gap in graphene by C–C bond alternation: a tight binding approach. *Materials Research Express*. 2019 Jan 9;6(4):045605.
11. Prasanna R, Gold-Parker A, Leijtens T, Conings B, Babayigit A, Boyen HG, Toney MF, McGehee MD. Band gap tuning via lattice contraction and octahedral tilting in perovskite materials for photovoltaics. *Journal of the American Chemical Society*. 2017 Aug 4;139(32):11117-24.
12. Peng X, Velasquez S. Strain modulated band gap of edge passivated armchair graphene nanoribbons. *Applied Physics Letters*. 2011 Jan 10;98(2):023112.

13. Park C, Atalla V, Smith S, Yoon M. Understanding the Charge Transfer at the Interface of Electron Donors and Acceptors: TTF–TCNQ as an Example. *ACS applied materials & interfaces*. 2017 Aug 2;9(32):27266-72.
14. Fernández AC, Castellani NJ. Dipole moment effects in dopamine/N-doped-graphene systems. *Surface Science*. 2020 Mar 1;693:121546.
15. Hybertsen MS, Louie SG. Electron correlation in semiconductors and insulators: Band gaps and quasiparticle energies. *Physical Review B*. 1986 Oct 15;34(8):5390.
16. Johnson KA, Ashcroft NW. Corrections to density-functional theory band gaps. *Physical Review B*. 1998 Dec 15;58(23):15548.

REVIEWERS' COMMENTS:

Reviewer #3 (Remarks to the Author):

Thanks for the authors' responses and revisions. I am satisfied with the current form of the manuscript and recommend publication.